# Optimal transport for variational data assimilation

Nelson Feyeux[1], Arthur Vidard[1], and Maëlle Nodet[1]

[1]Univ. Grenoble Alpes, Inria, CNRS, Grenoble INP, LJK, 38000 Grenoble, France

*Correspondence to:* A. Vidard (arthur.vidard@inria.fr)

**Abstract.**

Usually data assimilation methods evaluate observation-model misfits using weighted $L_2$ distances. However it is not well suited when observed features are present in the model with position error. In this context, the Wasserstein distance stemming from optimal transport theory is more relevant.

This paper proposes to adapt variational data assimilation to the use of such a measure. It provides a short introduction to optimal transport theory and discusses the importance of a proper choice of scalar product to compute the cost function gradient. It also extends the discussion to the way the descent is performed within the minimisation process.

These algorithmic changes are tested on a non-linear shallow-water model, leading to the conclusion that optimal transport-based data assimilation seems to be promising to capture position errors in the model trajectory.

10   *Copyright statement.*

## 1   Introduction

Understanding and forecasting the evolution of a given system is a crucial topic in an ever increasing number of application domains. To achieve this goal, one can rely on multiple sources of information, namely observations of the system, numerical model describing its behaviour, as well as additional *a priori* knowledge such as statistical information or previous forecasts. To combine these heterogeneous sources of observation it is common practice to use so-called data assimilation methods (*e.g.*, see reference books Lewis et al. (2006); Law et al. (2015); Asch et al. (2016)). Their aim is multiple: finding the initial and/or boundary conditions, parameter estimation, reanalysis, and so on. They are extensively used in numerical weather forecasting for instance (*e.g.*, see reviews in the books Park and Xu (2009, 2013)).

The estimation of the different elements to be sought, the control vector, is performed in data assimilation through the comparison between the observations and their model counterparts. The control vector should be adjusted such that its model outputs would fit the observations, while taking into account that these observations are imperfect and corrupted by noise and errors.

Data assimilation methods are divided into three distinct classes. First, there is statistical filtering based on Kalman filters. Then, variational data assimilation methods based on optimal control theory. More recently hybrids of both approaches have

been developed (Hamill and Snyder, 2000; Buehner, 2005; Bocquet and Sakov, 2014). In this paper we focus on variational data assimilation. It consists in minimizing a cost function written as the distance between the observations and their model counterparts. A Tikhonov regularization term is also added to the cost function as a distance between the control vector and a background state carrying a priori information.

Thus the cost function contains the misfit between the data (*a priori* and observations) and their control and model counterparts. Minimizing the cost function aims at reaching a compromise in which these errors are as small as possible. The errors can be decomposed into amplitude and position errors. Position errors mean that the structural elements are present in the data, but misplaced. Some methods have been proposed in order to deal with position errors (Hoffman and Grassotti, 1996; Ravela et al., 2007). These involve a preprocessing step which consists in displacing the different data so they fit better with each other.

Then the data assimilation is performed accounting for those displaced data.

A distance has to be chosen in order to compare the different data and measure the misfits. Usually, a Euclidean distance is used, often weighted to take into account the statistical errors. But Euclidean distances have trouble capturing position errors. This is illustrated in Fig. 1, which shows two curves $\rho_0$ and $\rho_1$. The second curve $\rho_1$ can be seen as the first one $\rho_0$ with position error. The minimizer of the cost function $\|\rho - \rho_0\|^2 + \|\rho - \rho_1\|^2$ is given by $\rho_* = \frac{1}{2}(\rho_0 + \rho_1)$, plotted with violet stars of Fig. 1. It

is the average of curves $\rho_0$ and $\rho_1$ with respect to the $\mathcal{L}^2$ distance. As we can see on Fig. 1, it does not correct for position error, but instead creates two smaller amplitude curves. We investigate in this article the idea of using instead a distance stemming from optimal transport theory, the Wasserstein distance, which can take into account position errors. In Fig. 1 we plot (green dots) the average of $\rho_0$ and $\rho_1$ with respect to the Wasserstein distance. Contrary to the $\mathcal{L}^2$ average, the Wasserstein average is what we want it to be: same shape, same amplitude, located in-between. It conserves the shape of the data. This is what we

want to achieve when dealing with position errors.

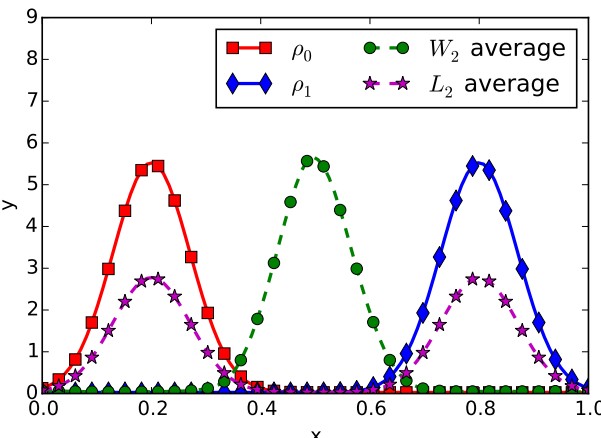

**Figure 1.** Wasserstein ($\mathcal{W}$) and Euclidean ($\mathcal{L}_2$) averages of two curves $\rho_0$ and $\rho_1$.

Optimal transport theory has been pioneered by Monge in 1781 (Monge, 1781). He searched for the optimal way of displacing sand piles onto holes of the same volume, minimizing the total cost of displacement. This can be seen as a transportation problem between two probability measures. A modern presentation can be found in Villani (2003) and will be discussed in Section 2.2.

Optimal transport has a wide spectrum of applications, from pure mathematical analysis on Riemannian spaces to applied economics, from functional inequalities (Cordero-Erausquin et al., 2004) to the semi-geostrophic equations (Cullen and Gangbo, 2001), through astrophysics (Brenier et al., 2003), medicine (Ratner et al., 2015), crowd motion (Maury et al., 2010) or urban planning (Buttazzo and Santambrogio, 2005). From optimal transport theory several distances can be derived, the most widely known being the Wasserstein distance (denoted $\mathcal{W}$) which is sensitive to misplaced features, and is the primary focus of this paper. This distance is also widely used in computer vision, for example in classification of images (Rubner et al., 1998, 2000), interpolation (Bonneel et al., 2011), or movie reconstruction (Delon and Desolneux, 2010). More recently, Farchi et al. (2016) used the Wasserstein distance to compare observation and model simulations in an air pollution context, which is a first step toward data assimilation.

Actual use of optimal transport in a variational data assimilation has been proposed by Ning et al. (2014), to tackle model error. The authors use the Wasserstein distance instead of the classical $\mathcal{L}^2$ norm *for model error control* in the cost function, and they offer promising results. Our contribution is in essence similar to them, in the fact that the Wasserstein distance is proposed in place of the $\mathcal{L}^2$ distance. Looking more closely, we investigate a different question, namely the idea of using the Wasserstein distance to measure *the observation misfit*. Also, we underline and investigate the impact of the choice of the scalar products, gradient formulations, as well as minimization algorithm choices on the assimilation performance, which is not discussed in Ning et al. (2014). These particularly subtle mathematical considerations are indeed crucial for the algorithm convergence, as will be shown in this paper, and is our main contribution.

The goal of the paper is to perform variational data assimilation with a cost function written with the Wasserstein distance. It may be extended to other type of data assimilation methods such as filtering methods, which largely exceeds the scope of this paper.

The present paper is organized as follows: first, in Section 2, variational data assimilation as well as Wasserstein distance are defined, and the ingredients required in the following are presented. The core of our contribution lies in Section 3: we first present the Wasserstein cost function, then propose two choices for its gradients, as well as two optimization strategies for the minimization. In Section 4 we present numerical illustrations, discuss the choices for the gradients and compare the optimization methods. Also, some difficulties related to the use of optimal transport will be pointed out and solutions will be proposed.

## 2 Materials and Methodology

This section deals with the presentation of variational data assimilation concepts and method on the one hand, and optimal transport and Wasserstein distance concepts, principles, and main theorems on the other hand. Section 3 will combine both worlds and will constitute the core of our original contribution.

### 2.1 Variational data assimilation

This paper focuses on variational data assimilation in the framework of initial state estimation. Let us assume that a system state is described by a variable $\mathbf{x}$, denoted $\mathbf{x}_0$ at initial time. We are also given observations $\mathbf{y}^{\text{obs}}$ of the system, which might be indirect, incomplete and approximate. Initial state and the observations are linked by operator $\mathcal{G}$, mapping the system initial state $\mathbf{x}_0$ to the observation space, so that $\mathcal{G}(\mathbf{x}_0)$ and $\mathbf{y}^{\text{obs}}$ belong to the same space. Usually $\mathcal{G}$ is defined using two other operators, namely the model $\mathcal{M}$ which gives the model state as a function of the initial state and the observation operator $\mathcal{H}$ which maps the system state to the observation space, such that $\mathcal{G} = \mathcal{H} \circ \mathcal{M}$.

Data assimilation aims to find a good estimate of $\mathbf{x}_0$ using the observations $\mathbf{y}^{\text{obs}}$ and the knowledge of the operator $\mathcal{G}$. Variational data assimilation methods do so by finding the minimizer $\mathbf{x}_0$ of the misfit function $\mathcal{J}$ (the cost function) between the observations $\mathbf{y}^{\text{obs}}$ and their computed counterparts $\mathcal{G}(\mathbf{x}_0)$,

$$\mathcal{J}(\mathbf{x}_0) = d_R(\mathcal{G}(\mathbf{x}_0), \mathbf{y}^{\text{obs}})^2$$

with $d_R$ some distance to be defined. Generally, this problem is ill-posed. For the minimizer of $\mathcal{J}$ to be unique, a background term is added and acts like a Tikhonov regularization. This background term is generally expressed as the distance with a background term $\mathbf{x}^b$, which contains *a priori* informations. The actual cost function then reads

$$\mathcal{J}(\mathbf{x}_0) = d_R(\mathcal{G}(\mathbf{x}_0), \mathbf{y}^{\text{obs}})^2 + d_B(\mathbf{x}_0, \mathbf{x}^b)^2, \tag{1}$$

with $d_B$ another distance to be specified. The control of $\mathbf{x}_0$ is done by the minimization of $\mathcal{J}$. Such minimization is generally carried out numerically using gradient descent methods. Paragraph 3.3 will give more details about the minimization process.

The distances to the observations $d_R$ and to the background term $d_B$ have to be chosen in this formulation. Usually, Euclidean distances ($\mathcal{L}^2$ distances, potentially weighted) are chosen, giving the following Euclidean cost function

$$\mathcal{J}(\mathbf{x}_0) = \|\mathcal{G}(\mathbf{x}_0) - \mathbf{y}^{\text{obs}}\|_2^2 + \|\mathbf{x}_0 - \mathbf{x}^b\|_2^2, \tag{2}$$

with $\|\cdot\|_2$ the $\mathcal{L}^2$ norm defined by

$$\|\mathbf{a}\|_2^2 := \int |\mathbf{a}(x)|^2 \, \mathrm{d}x. \tag{3}$$

Euclidean distances, such as the $\mathcal{L}^2$ distance, are local metrics. In the following we will investigate the use of a non-local metric, the Wasserstein distance $\mathcal{W}$, in place of $d_R$ and $d_B$ in equation (1). Such a cost function will be presented in Section 3. The Wasserstein distance is presented and defined in the following subsection.

## 2.2 Optimal transport and Wasserstein distance

The essentials of optimal transport theory and Wasserstein distance required for data assimilation are presented.

We define, in this order, the space of mass functions where the Wasserstein distance is defined, then the Wasserstein distance and finally the Wasserstein scalar product, a key ingredient for variational assimilation.

### 2.2.1 Mass functions

We consider the case where the observations can be represented as positive fields that we will call "mass functions". A mass function is a non-negative function of space. For example, a grey-scaled image is a mass function, it can be seen as a function of space to the interval $[0,1]$ where 0 encodes black and 1 encodes white.

**Definition 2.1.** Let $\Omega$ be a closed, convex, bounded set of $\mathbb{R}^d$ and let define the set of mass functions $\mathcal{P}(\Omega)$ be the set of non-negative functions of total mass 1:

$$\mathcal{P}(\Omega) := \left\{ \rho \geq 0 \colon \int_\Omega \rho(x)\,\mathrm{d}x = 1 \right\}. \tag{4}$$

Let us remark here that, in the mathematical framework of optimal transport, mass functions are continuous and they are called "probability densities". In the data assimilation framework the concept of probability densities is mostly used to represent errors. Here, the positive functions we consider actually serve as *observations* or *state vectors*, so we chose to call them *mass functions* to avoid any possible confusion with state or observation error probability distributions.

### 2.2.2 Wasserstein distance

The optimal transport problem is to compute among all the transportations between two mass functions, the one minimizing the kinetic energy. A transportation between two mass functions $\rho_0$ and $\rho_1$ is given by a time path $\rho(t,x)$ such that $\rho(t=0) = \rho_0$ and $\rho(t=1) = \rho_1$, and a velocity field $\mathbf{v}(t,x)$ such that the continuity equation holds,

$$\frac{\partial \rho}{\partial t} + \mathrm{div}(\rho \mathbf{v}) = 0. \tag{5}$$

Such a path $\rho(t)$ can be seen as interpolating $\rho_0$ and $\rho_1$. For $\rho(t)$ to stay in $\mathcal{P}(\Omega)$, a sufficient condition is that the velocity field $\mathbf{v}(t,x)$ should be tangent to the domain boundary, meaning that $\rho(t,x)\mathbf{v}(t,x)\cdot\boldsymbol{n}(x) = 0$ for almost all $(t,x) \in [0,1] \times \partial\Omega$. With this condition, the support of $\rho(t)$ remains in $\Omega$.

Let us be clear here that the time $t$ is fictitious, and has no relationship whatsoever with the physical time of data assimilation. It is purely used to define the Wasserstein distance and some mathematically related objects.

The Wasserstein distance $\mathcal{W}$ is hence the minimum in terms of kinetic energy among all the transportations between $\rho_0$ and $\rho_1$,

$$\mathcal{W}(\rho_0,\rho_1) = \sqrt{\min_{(\rho,\mathbf{v})\in C(\rho_0,\rho_1)} \iint_{[0,1]\times\Omega} \rho(t,x)|\mathbf{v}(t,x)|^2\,\mathrm{d}t\mathrm{d}x} \tag{6}$$

with $C(\rho_0, \rho_1)$ representing the set of continuous transportations between $\rho_0$ and $\rho_1$ described by a velocity field $\mathbf{v}$ tangent to the boundary of the domain,

$$C(\rho_0, \rho_1) := \left\{ (\rho, \mathbf{v}) \text{ s.t. } \begin{array}{l} \partial_t \rho + \text{div}(\rho \mathbf{v}) = 0, \\ \rho(t=0) = \rho_0, \ \rho(t=1) = \rho_1, \\ \rho \mathbf{v} \cdot \boldsymbol{n} = 0 \text{ on } \partial\Omega \end{array} \right\}. \tag{7}$$

This definition of the Wasserstein distance is the Benamou-Brenier formulation (Benamou and Brenier, 2000). There exist other definitions, based on the transport map or the transference plans, but slightly out of the scope of this article. See the introduction of Villani (2003) for more details.

A remarkable property is that the optimal velocity field $\mathbf{v}$ is of the form

$$\mathbf{v}(t, x) = \nabla \Phi(t, x)$$

with $\Phi$ following the Hamilton-Jacobi equation (Benamou and Brenier, 2000)

$$\partial_t \Phi + \frac{|\nabla \Phi|^2}{2} = 0. \tag{8}$$

The equation of the optimal $\rho$ is the continuity equation using this velocity field. Moreover, the function $\Psi$ defined by

$$\Psi(x) := -\Phi(t=0, x) \tag{9}$$

is said to be the **Kantorovich potential** of the transport between $\rho_0$ and $\rho_1$. It is a useful feature in the derivation of the Wasserstein cost function presented in Section 3.

A remarkable property of the Kantorovich potential allows to compute the Wasserstein distance, this is the Benamou-Brenier formula (see (Benamou and Brenier, 2000) or (Villani, 2003, Th. 8.1)), given by

$$\mathcal{W}(\rho_0, \rho_1)^2 = \int_\Omega \rho_0(x) |\nabla \Psi(x)|^2 \, dx \tag{10}$$

**Example 2.2.** The classical example for optimal transport is the transport of Gaussian mass functions. For $\Omega = \mathbb{R}^d$, let us consider two Gaussian mass functions: $\rho_i$ of mean $\mu_i$ and variance $\sigma_i^2$ for $i = 0$ and $i = 1$. Then the optimal transport $\rho(t)$ between $\rho_0$ and $\rho_1$ is a transportation-dilatation function of $\rho_0$ to $\rho_1$. More precisely, $\rho(t)$ is a Gaussian mass function whose mean is $\mu_0 + t(\mu_1 - \mu_0)$ and variance is $(\sigma_0 + t(\sigma_1 - \sigma_0))^2$. The corresponding computed Kantorovich potential is (up to a constant):

$$\Psi(x) = \left( \frac{\sigma_1}{\sigma_0} - 1 \right) \frac{|x|^2}{2} + \left( \mu_1 - \frac{\sigma_1}{\sigma_0} \mu_0 \right) \cdot x.$$

Finally, a few words should be said about the numerical computation of the Wasserstein distance. In one dimension, the optimal transport $\rho(t, x)$ is easy to compute as the Kantorovich potential has an exact formulation: the Kantorovich potential of the transport between two mass functions $\rho_0$ and $\rho_1$ is the only function $\Psi$ such that

$$F_1(x - \nabla \Psi(x)) = F_0(x), \quad \forall x \tag{11}$$

with $F_i$ the cumulative distribution function of $\rho_i$. Numerically we fix $x$ and solve iteratively equation (11) using binary search to find $\nabla\Psi$. Then, we obtain $\Psi$ thanks to numerical integration. Finally, equation (10) gives the Wasserstein distance.

For two or three dimensional problems, there exists no general formula for the Wasserstein distance and more complex algorithms have to be used, like the (iterative) primal-dual one (Papadakis et al., 2014) or the semi-discrete one (Mérigot, 2011). In the former, an approximation of the Kantorovich potential is directly read in the so-called dual variable.

### 2.2.3 Wasserstein inner product

The scalar product between two functions is required for data assimilation and optimization: as we will recall later, the scalar product choice is used to define the gradient value. This paper will consider the classical $\mathcal{L}^2$ scalar product as well as the one associated to the Wasserstein distance. A scalar product defines the angle and norm of vectors tangent to $\mathcal{P}(\Omega)$ at a point $\rho_0$. First, a tangent vector in $\rho_0$ is the derivative of a curve $\rho(t)$ passing through $\rho_0$. As a curve $\rho(t)$ can be described by a continuity equation, the space of tangent vectors, the tangent space, is formally defined by (cf. Otto, 2001),

$$T_{\rho_0}\mathcal{P} = \left\{ \eta \in \mathcal{L}^2(\Omega), \text{ s.t. } \eta = -\operatorname{div}(\rho_0 \nabla\Phi) \text{ with } \Phi \text{ s.t. } \rho_0 \frac{\partial\Phi}{\partial\boldsymbol{n}} = 0 \text{ on } \partial\Omega \right\}. \tag{12}$$

Let us first recall that the Euclidean, or $\mathcal{L}^2$, scalar product $\langle\cdot,\cdot\rangle_2$ is defined on $T_{\rho_0}\mathcal{P}$ by

$$\forall \eta, \eta' \in T_{\rho_0}\mathcal{P}(\Omega), \quad \langle\eta,\eta'\rangle_2 := \int_\Omega \eta(x)\eta'(x)\,\mathrm{d}x. \tag{13}$$

The Wasserstein inner product $\langle\cdot,\cdot\rangle_W$ is defined for $\eta = -\operatorname{div}(\rho_0\nabla\Phi), \eta' = -\operatorname{div}(\rho_0\nabla\Phi') \in T_{\rho_0}\mathcal{P}$ by

$$\langle\eta,\eta'\rangle_W := \int_\Omega \rho_0 \nabla\Phi \cdot \nabla\Phi'\,\mathrm{d}x. \tag{14}$$

One has to note that the inner product is dependent on $\rho_0 \in \mathcal{P}(\Omega)$. Finally, the norm associated to a tangent vector $\eta = -\operatorname{div}(\rho_0\nabla\Phi) \in T_{\rho_0}\mathcal{P}$ is

$$\|\eta\|_W^2 = \int_\Omega \rho_0 |\nabla\Phi|^2\,\mathrm{d}x \tag{15}$$

hence the kinetic energy of the small displacement $\eta$. This point makes the link between this inner product and the Wasserstein distance.

## 3 Optimal transport-based data assimilation

This section is our main contribution. First, we will consider the Wasserstein distance to compute the *observation term of the cost function*; second, we will discuss the choices of the scalar product and the gradient descent method, and their impact on the assimilation algorithm efficiency.

## 3.1 Wasserstein cost function

In the framework of Section 2.2 we will define the data assimilation cost function using the Wasserstein distance. For this cost function to be well defined we assume that the control variables belong to $\mathcal{P}(\Omega)$ and that the observation variables belong to another space $\mathcal{P}(\Omega_o)$ with $\Omega_o$ a closed, convex, bounded set of $\mathbb{R}^{d'}$. Let us recall that this means that they are all non-negative functions with integral equal to 1. Having elements with integral 1 (or constant integral) may seem restrictive. Removing it is possible by using a modified version of the Wasserstein distance, presented for example in Chizat et al. (2015) or Farchi et al. (2016). For simplicity we do not consider this possible generalization and all data have the same integral. The cost function (1) is rewritten using the Wasserstein distance defined in Section 2.2,

$$\mathcal{J}_{\mathcal{W}}(\mathbf{x}_0) = \frac{1}{2} \sum_{i=1}^{N^{\mathrm{obs}}} \mathcal{W}(\mathcal{G}_i(\mathbf{x}_0), \mathbf{y}_i^{\mathrm{obs}})^2 + \frac{\omega_b}{2} \mathcal{W}(\mathbf{x}_0, \mathbf{x}_0^b)^2 \tag{16}$$

with $\mathcal{G}_i : \mathcal{P}(\Omega) \to \mathcal{P}(\Omega_o)$ the observation operator computing the $\mathbf{y}_i^{\mathrm{obs}}$ counterpart from $\mathbf{x}_0$ and $\omega_b$ a scalar weight associated to the background term.

The variables $\mathbf{x}_0$ and $\mathbf{y}_i^{obs}$ may be vectors whose components are functions belonging to $\mathcal{P}(\Omega)$ and $\mathcal{P}(\Omega_o)$, respectively. The Wasserstein distance between two such vectors is the sum of the distances between their components. The remainder of the article is easily adaptable to this case, but for simplicity we set $\mathbf{x}_0 = \rho_0 \in \mathcal{P}(\Omega)$ and $\mathbf{y}_i^{obs} = \rho_i^{\mathrm{obs}} \in \mathcal{P}(\Omega)$. The Wasserstein cost function (16) then becomes

$$\mathcal{J}_{\mathcal{W}}(\rho_0) = \frac{1}{2} \sum_{i=1}^{N^{\mathrm{obs}}} \mathcal{W}(\mathcal{G}_i(\rho_0), \rho_i^{\mathrm{obs}})^2 \quad + \quad \frac{\omega_b}{2} \mathcal{W}(\rho_0, \rho_0^b)^2. \tag{17}$$

As for the classical $\mathcal{L}^2$ cost function, $\mathcal{J}_{\mathcal{W}}$ is convex with respect to the Wasserstein distance in the linear case, and has a unique minimizer. In the non-linear case, the uniqueness of the minimizer relies on the regularization term $\frac{\omega_b}{2} \mathcal{W}(\rho_0, \rho_0^b)^2$.

To find the minimum of $\mathcal{J}_{\mathcal{W}}$, a gradient descent method is applied. It is presented in Section 3.3. As this type of algorithm requires the gradient of the cost function, computation of the gradient of $\mathcal{J}_{\mathcal{W}}$ is the focus of next Section.

## 3.2 Gradient of $\mathcal{J}_{\mathcal{W}}$

If $\mathcal{J}_{\mathcal{W}}$ is differentiable, its gradient is given by

$$\forall \eta \in T_{\rho_0}\mathcal{P}, \quad \lim_{\epsilon \to 0} \frac{\mathcal{J}_{\mathcal{W}}(\rho_0 + \epsilon\eta) - \mathcal{J}_{\mathcal{W}}(\rho_0)}{\epsilon} = \langle \eta, g \rangle \tag{18}$$

where $\langle \cdot, \cdot \rangle$ represents the scalar product. The scalar product is not unique, so as a consequence neither is the gradient. In this work we decided to study and compare two choices for the scalar product, the natural one $\mathcal{W}$ and the usual one $\mathcal{L}^2$. $\mathcal{W}$ is clearly the ideal candidate for a good scalar product. However, we also decided to study the $\mathcal{L}^2$ scalar product because it is the usual choice in optimisation. Numerical comparison is done in Section 4.

The associated gradients are respectively denoted $\mathrm{grad}_W \mathcal{J}_W(\rho_0)$ and $\mathrm{grad}_2 \mathcal{J}_W(\rho_0)$ and are the only elements of the tangent space $T_{\rho_0}\mathcal{P}$ of $\rho_0 \in \mathcal{P}(\Omega)$ such that

$$\forall \eta \in T_{\rho_0}\mathcal{P}, \quad \lim_{\epsilon \to 0} \frac{\mathcal{J}_W(\rho_0 + \epsilon\eta) - \mathcal{J}_W(\rho_0)}{\epsilon} = \langle \mathrm{grad}_W \mathcal{J}_W(\rho_0), \eta \rangle_W$$

$$= \langle \mathrm{grad}_2 \mathcal{J}_W(\rho_0), \eta \rangle_2. \tag{19}$$

Here in the notations, the word "grad" is used for the gradient of a function while the spatial gradient is denoted by the nabla sign $\nabla$. The gradients of $\mathcal{J}_W$ are elements of $T_{\rho_0}\mathcal{P}$ and hence functions of space.

The following theorem allows to compute both gradients of $\mathcal{J}_W$:

**Theorem 3.1.** *For $i \in \{1,\ldots,N^{obs}\}$, let $\Psi^i$ be the Kantorovich potential (see equation (9)) of the transport between $\mathcal{G}_i(\rho_0)$ and $\rho_i^{obs}$. Let $\Psi^b$ be the Kantorovich potential of the transport map between $\rho_0$ and $\rho_0^b$. Then,*

$$\mathrm{grad}_2 \mathcal{J}_W(\rho_0) = \omega_b \Psi^b + \sum_{i=1}^{N^{obs}} \mathbf{G}_i^*(\rho_0).\Psi^i + c \tag{20}$$

*with $c$ such that the integral of $\mathrm{grad}_2 \mathcal{J}_W(\rho_0)$ is zero, and $\mathbf{G}_i^*$ the adjoint of $\mathcal{G}_i$ w.r.t. the $\mathcal{L}_2$ inner product (see definition reminder below). Assuming that $\mathrm{grad}_2 \mathcal{J}_W(\rho_0)$ has the no-flux boundary condition (see comment about this assumption below)*

$$\rho_0 \frac{\partial \mathrm{grad}_2 \mathcal{J}_W(\rho_0)}{\partial \boldsymbol{n}} = 0 \text{ on } \partial\Omega$$

*then the gradient w.r.t. the Wasserstein inner product is*

$$\mathrm{grad}_W \mathcal{J}_W(\rho_0) = -\mathrm{div}\left(\rho_0 \nabla[\mathrm{grad}_2 \mathcal{J}_W(\rho_0)]\right). \tag{21}$$

(A proof of this Theorem can be found in Appendix A.)

The adjoint $\mathbf{G}_i^*(\rho_0)$ is defined by the classical equality

$$\forall \eta, \mu \in T_{\rho_0}\mathcal{P}, \langle \mathbf{G}_i^*(\rho_0).\mu, \eta \rangle_2 = \langle \mu, \mathbf{G}_i(\rho_0).\eta \rangle_2 \tag{22}$$

where $\mathbf{G}_i[\rho_0]$ is the tangent model, defined by

$$\forall \eta \in T_{\rho_0}\mathcal{P}, \mathbf{G}_i(\rho_0).\eta := \lim_{\epsilon \to 0} \frac{\mathcal{G}_i(\rho_0 + \epsilon\eta) - \mathcal{G}_i(\rho_0)}{\epsilon}. \tag{23}$$

Note that the *no-flux boundary condition assumption* for $\mathrm{grad}_2 \mathcal{J}_W(\rho_0)$, that is

$$\rho_0 \frac{\partial \mathrm{grad}_2 \mathcal{J}_W(\rho_0)}{\partial \boldsymbol{n}} = 0 \text{ on } \partial\Omega$$

is not necessarily satisfied. The Kantorovich potentials respect this condition. Indeed, their spatial gradients are velocities thus tangent to the boundary, see the end of Section 2.2. But it may not be conserved through the mapping with the adjoint model, $\mathbf{G}_i^*(\rho_0)$. In the case where $\mathbf{G}_i^*(\rho_0)$ does not preserve this condition, the Wasserstein gradient is not of integral zero. A possible workaround is to use a product coming from the unbalanced Wasserstein distance of Chizat et al. (2015).

## 3.3 Minimization of $\mathcal{J}_\mathcal{W}$

The minimizer of $\mathcal{J}_\mathcal{W}$ defined in (17) is expected to be a good trade-off between both the observations and the background with respect to the Wasserstein distance and to have good properties, as shown in Fig. 1. It can be computed through an iterative gradient-based descent method. Such methods start from a control state $\rho_0^0$ and step-by-step update it using an iteration of the

form

$$\rho_0^{n+1} = \rho_0^n - \alpha^n d^n \tag{24}$$

where $\alpha^n$ is a real number (the step) and $d^n$ is a function (the descent direction), chosen such that $\mathcal{J}_\mathcal{W}(\rho_0^{n+1}) < \mathcal{J}_\mathcal{W}(\rho_0^n)$. In gradient-based descent methods, $d^n$ can be equal to the gradient of $\mathcal{J}_\mathcal{W}$ (steepest descent method), or to a function of the gradient and $d^{n-1}$ (conjugate gradient, quasi-Newton methods, ...). Under sufficient conditions on $(\alpha^n)$, the sequence $(\rho_0^n)$

converges to a local minimizer. See Nocedal and Wright (2006) for more details.

We will now explain how to adapt the gradient descent to the optimal transport framework. With the Wasserstein gradient (21), the descent of $\mathcal{J}_\mathcal{W}$ follows an iteration scheme of the form

$$\rho_0^{n+1} = \rho_0^n + \alpha^n \operatorname{div}(\rho_0^n \nabla \Phi^n) \tag{25}$$

with $\alpha^n > 0$ to be chosen.

The inconveniences of this iteration are twofold. First, for $\rho_0^{n+1}$ to be non-negative, $\alpha^n$ may have to be very small. Second, the supports of functions $\rho_0^{n+1}$ and $\rho_0^n$ are the same. A more *transport-like* iteration could be used instead, by making $\rho_0^n$ follow the geodesics in the Wasserstein space. All geodesics $\rho(\alpha)$ starting from $\rho_0^n$ are solutions of the set of partial differential equations

$$\begin{cases} \partial_\alpha \rho + \operatorname{div}(\rho \nabla \Phi) = 0, \qquad \rho(\alpha = 0) = \rho_0^n, \\ \partial_\alpha \Phi + \dfrac{|\nabla \Phi|^2}{2} = 0, \end{cases} \tag{26}$$

see equation (8). Furthermore, two different values of $\Phi(\alpha = 0)$ give two different geodesics. In the optimal transport theory community, the geodesic $\rho(\alpha)$ starting from $\rho_0^n$ with initial condition $\Phi(\alpha = 0) = \Phi_0$ would be written with the following notation:

$$\rho(\alpha) = (I - \alpha \nabla \Phi_0) \# \rho_0^n. \tag{27}$$

see (Villani, 2003, Section 8.2) for more details.

For the gradient iteration, we choose the geodesic starting from $\rho_0^n$ with initial condition $\Phi(\alpha = 0) = \Phi^n$, i.e. using the optimal transport notation $\rho_0^{n+1}$ is given by

$$\rho_0^{n+1} = (I - \alpha^n \nabla \Phi^n) \# \rho_0^n \tag{28}$$

with $\alpha^n > 0$ to be chosen. This descent is consistent with (25) because (25) is the first order discretization of (26) with $\Phi(\alpha = 0) = \Phi^n$. Therefore, (28) and (25) are equivalent when $\alpha^n \to 0$.

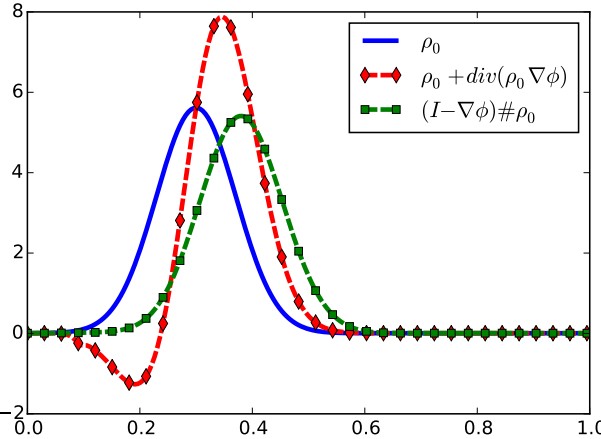

**Figure 2.** Comparison of iterations (25) and (28) with $\rho_0$ of limited support and $\Phi$ such that $\nabla\Phi$ is constant on the support of $\rho_0$.

Comparison of (28) and (25) is shown on Fig. 2 for simple $\rho_0^n$ and $\Phi$. This comparison depicts the usual advantage of using (28) instead of (25): the former is always in $\mathcal{P}(\Omega)$ and supports of functions change. Iteration (28) is the one used in the following numerical experiments.

## 4  Numerical illustrations

Let us recall that in the data assimilation vocabulary, the word "analysis" refers to the minimizer of the cost function at the end of the data assimilation process.

In this section the analyses resulting from the minimization of the Wasserstein cost function defined previously in (16) are presented, in particular when position errors occur. Results are compared with the results given by the $\mathcal{L}^2$ cost function defined in (2).

The experiments are all one-dimensional and $\Omega$ is the interval $[0,1]$. A discretization of $\Omega$ is performed and involves 200 uniformly distributed discretization points. A first, simple experiment uses a linear operator $\mathcal{G}$. In a second experiment, the operator is non-linear.

Only a single variable is controlled. This variable $\rho_0$ represents the initial condition of an evolution problem. It is an element of $\mathcal{P}(\Omega)$, and observations are also elements of $\mathcal{P}(\Omega)$.

In this paper we chose to work in the twin experiments framework. In this context the true state, denoted $\rho_0^t$, is known and used to generate the observations: $\rho_i^{\text{obs}} = \mathcal{G}_i(\rho_0^t)$ at various times $(t_i)_{i=1..N_{\text{obs}}}$. Observations are first perfect, that is noise-free and available everywhere in space. Then in Section 4.3, we will add noise in the observations. The background term is supposed to have position errors only and no amplitude error. The data assimilation process aims to recover a good estimation of the

true state, using the cost function involving the simulated observations and the background term. The analysis obtained after convergence can then be compared to the true state and effectiveness diagnostics can be made.

Both the Wasserstein (17) and $\mathcal{L}^2$ (2) cost functions are minimized through a steepest gradient method. The $\mathcal{L}^2$ gradient is used to minimize the $\mathcal{L}^2$ cost function. Both the $\mathcal{L}^2$ and $\mathcal{W}$ gradients are used for the Wasserstein cost functions (cf. Theorem 3.1 for expressions of both gradients), giving respectively, with $\Phi^n := \mathrm{grad}_2 \mathcal{J}_{\mathcal{W}}(\rho_0^n)$, the iterations

$$\rho_0^{n+1} = \rho_0^n - \alpha^n \Phi^n \tag{DG2}$$

$$\rho_0^{n+1} = (I - \alpha^n \nabla \Phi^n) \# \rho_0^n. \tag{DG\#}$$

The value of $\alpha^n$ is chosen close to optimal using a line search algorithm and the descent stops when the decrement of $\mathcal{J}$ between two iterations is lower than $10^{-6}$.

## 4.1 Linear example

The first example involves a linear evolution model as $(\mathcal{G}_i)_{i=1..N_{\mathrm{obs}}}$ with the number of observations $N_{\mathrm{obs}}$ equal to 5. Every single operator $\mathcal{G}_i$ maps an initial condition $\rho_0$ to $\rho(t_i)$ according to the following continuity equation defined in $\Omega = [0,1]$,

$$\partial_t \rho + u \cdot \nabla \rho = 0 \text{ with } u = 1. \tag{30}$$

The operator $\mathcal{G}_i$ is linear. We control $\rho_0$ only. The true state $\rho_0^t \in \mathcal{P}(\Omega)$ is a localised mass function, similar to the background term $\rho_0^b$ but located at a different place, as if it had position errors. The true and background states as well as the observations at various times are plotted on Fig. 3 (top). The computed analysis $\rho_0^{a,2}$ for the $\mathcal{L}_2$ cost function is shown on Fig. 3 (bottom left). This Figure shows also the analysis $\rho_0^{a,W}$ corresponding to both (DG2) and (DG#) algorithms minimizing the same Wasserstein $\mathcal{J}_{\mathcal{W}}$ cost function.

As expected in the introduction, see *e.g.* Fig. 1, minimizing $\mathcal{J}_2$ leads to an analysis $\rho_0^{a,2}$ being the $\mathcal{L}^2$-average of the background and true states (hence two small localised mass functions), while $\mathcal{J}_{\mathcal{W}}$ leads to a satisfactorily shaped analysis $\rho_0^{a,W}$ in-between the background and true states.

The issue of amplitude of the analysis of $\rho_0^{a,2}$ and the issue of position of $\rho_0^{a,W}$ are not corrected by the time evolution of the model, as shows Fig. 3 (bottom right). At the end of the assimilation window, each of both of the analyses still have discrepancies with the observations.

Both of the algorithms (DG2) and (DG#) give the same analysis, the minimum of $\mathcal{J}_{\mathcal{W}}$. However, the convergence speed is not the same at all. The values of $\mathcal{J}_{\mathcal{W}}$ throughout the algorithm are plotted on Fig. 4. It can be seen that (DG#) converges in a couple of iterations while (DG2) needs more than 2000 iterations to converge. It is a very slow algorithm because it does not provide the steepest descent associated to the Wasserstein metric. The Figure also shows that even in a conjugate gradient (CG) version of (DG2), the descent is still quite slow (it needs $\sim 100$ iterations to converge). This comparison highlights the need for a well-suited inner product and more precisely that the $\mathcal{L}^2$ one is not fit for the Wasserstein distance.

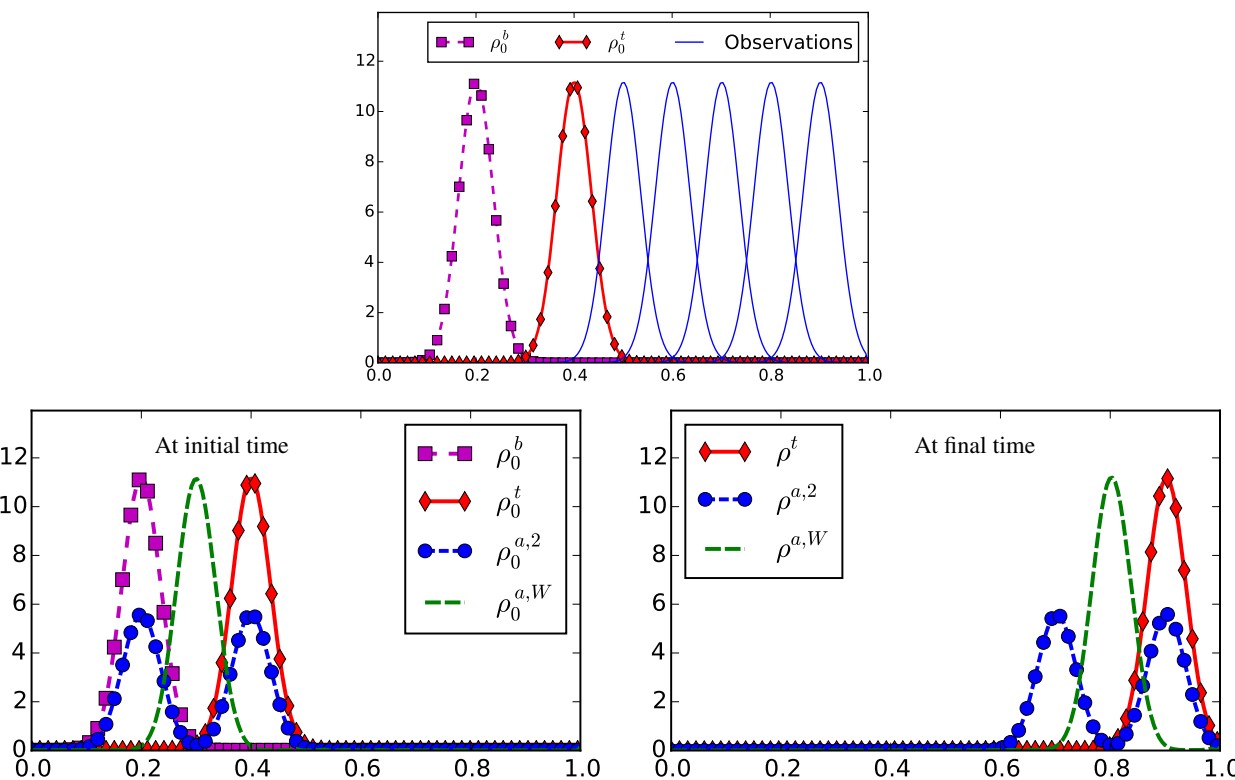

**Figure 3.** Top: the twin experiments ingredients are plotted, namely true initial condition $\rho_0^t$, background term $\rho_0^b$, and observations at different times. Bottom left: we plot the analyses obtained after each proposed method, compared to $\rho_0^b$ and $\rho_0^t$: $\rho_0^{a,2}$ corresponds to $\mathcal{J}_2$ while $\rho_0^{a,W}$ to both (DG2) and (DG#). Bottom right: fields at final time, $\rho^t$, $\rho^{a,2}$ and $\rho^{a,W}$, when taking respectively $\rho_0^t$, $\rho_0^{a,2}$ and $\rho_0^{a,W}$ as initial condition.

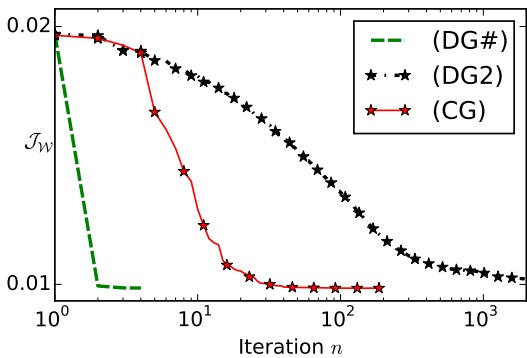

**Figure 4.** Decreasing of $\mathcal{J}_{\mathcal{W}}$ through the iterations of (DG#) and (DG2), and a conjugate gradient version (CG) of (DG2).

As a conclusion of this first test case, we managed to write and minimize a cost function which gives a relevant analysis, contrary to what we obtain with the classical Euclidean cost function, in case of position errors. We also noticed that the success of the minimization of $\mathcal{J}_{\mathcal{W}}$ was clearly dependent on the scalar product choice.

## 4.2 Non-linear example

Further results are shown when a non-linear model is used in place of $\mathcal{G}$. The framework and procedure are the same as the first test case, see the beginning of Section 4 and Section 4.1 for details. The non-linear model used is the shallow-water system described by

$$\begin{cases} \partial_t h + \partial_x(hu) = 0 \\ \partial_t u + u\partial_x u + g\partial_x h = 0 \end{cases}$$

subject to initial conditions $h(0) = h_0$ and $u(0) = u_0$, with reflective boundary conditions ($u|_{\partial\Omega} = 0$), where the constant $g$ is the gravity acceleration. The variable $h$ represents the water surface elevation, and $u$ is the current velocity. If $h_0$ belongs to $\mathcal{P}(\Omega)$, then the corresponding solution $h(t)$ belongs to $\mathcal{P}(\Omega)$.

The true state is $(h_0^t, u_0^t)$, where velocity $u_0^t$ is equal to 0 and surface elevation $h_0^t$ is a given localised mass function. The initial velocity field is supposed to be known and therefore not included in the control vector. Only $h_0$ is controlled, using $N_{\mathrm{obs}} = 5$ direct observations of $h$ and a background term $h_0^b$, also a localised mass function like $h_0^t$.

Data assimilation is performed by minimizing either the $\mathcal{J}_2$ or the $\mathcal{J}_{\mathcal{W}}$ cost functions described above. Thanks to the experience gained during the first experiment, only (DG#) algorithm is used for the minimization of $\mathcal{J}_{\mathcal{W}}$.

In Fig. 5 (top) we present initial surface elevation $h_0^t$, $h_0^b$ as well as 2 of the 10 observations used for the experiment. In Fig. 5 (bottom left), the analyses corresponding to $\mathcal{J}_2$ and $\mathcal{J}_{\mathcal{W}}$ are shown: $h_0^{a,2}$ and $h_0^{a,W,\#}$. Analysis $h_0^{a,2}$ is close to the $\mathcal{L}^2$-average of the true and background states, even at time $t > 0$, while $h_0^{a,W,\#}$ lies close to the Wasserstein-average between the background and true states, and hence has the same shape as them (see Fig. 1).

Figure 5 (bottom right) shows that at the end of the assimilation window, the surface elevation $h^{a,W,\#} = \mathcal{G}(h_0^{a,W,\#})$ is still more realistic than $h^{a,2} = \mathcal{G}(h_0^{a,2})$, when compared to the true state $h^t = \mathcal{G}(h_0^t)$.

The conclusion of this second test case is that even with non-linear models, our Wasserstein-based algorithm can give interesting results in case of position errors.

## 4.3 Robustness to observation noise

In this section, a noise in position and shape has been added in the observations. This type of noise typically occurs in images from satellites. For example, Fig. 6 (top) shows an observation from the previous experiment where peaks have been displaced and resized randomly. For each structure of each observations, the displacements and amplitude changes are independent and uncorrelated. This perturbation is done so that the total mass is preserved.

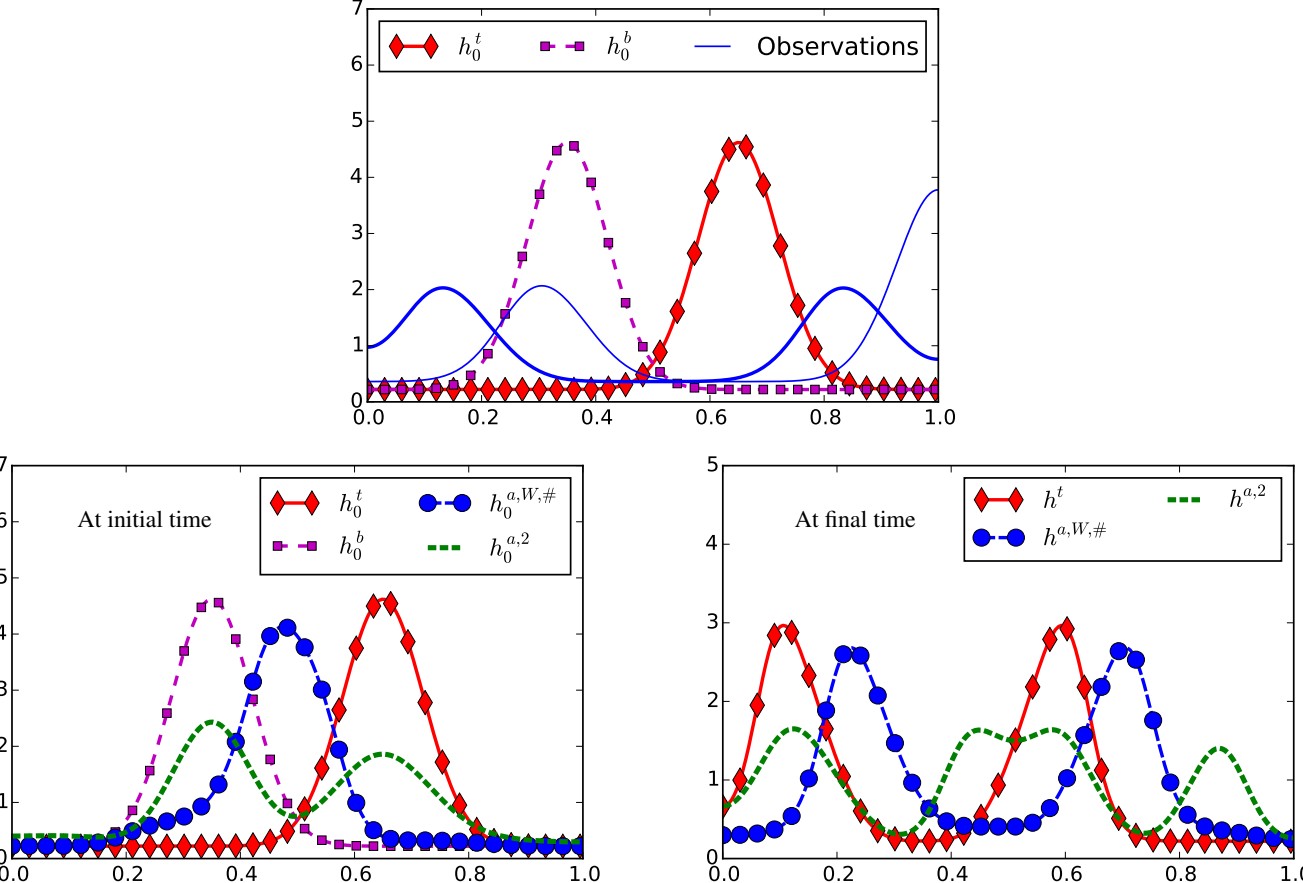

**Figure 5.** Top: Ingredients of the second experiment: true initial condition $h_0^t$, background $h_0^b$ and 2 of the 10 observations at different times. Bottom left: the true and background initial conditions are shown, and also the analyses $h_0^{a,2}$ and $h_0^{a,W}$ corresponding respectively to the Euclidean and Wasserstein cost functions. On the right we show the same plots (except the background one) but at the end of the assimilation window.

Analyses of this noisy experiment using $\mathcal{L}^2$ (1) and Wasserstein (17) cost functions are compared to analyses from the last experiment where no noise was present.

For the $\mathcal{L}^2$ cost function, surface elevation analyses $h_0^{a,2}$ are shown in Fig. 6 (bottom left). We see that adding such a noise in the observations degrades the analysis. In particular, the right peak (associated to the observations) is more widely spread: this is a consequence of the fact that the $\mathcal{L}^2$ distance is a local-in-space distance.

For the Wasserstein cost function, analyses $h_0^{a,W}$ are shown in Fig. 6 (bottom right). The analysis does not change much with the presence of noise and remains similar to the one obtained in the previous experiment. This is a consequence of a property of the Wasserstein distance: the Wasserstein barycenter of several Gaussians is a Gaussian with averaged position and variance (see example 2.2).

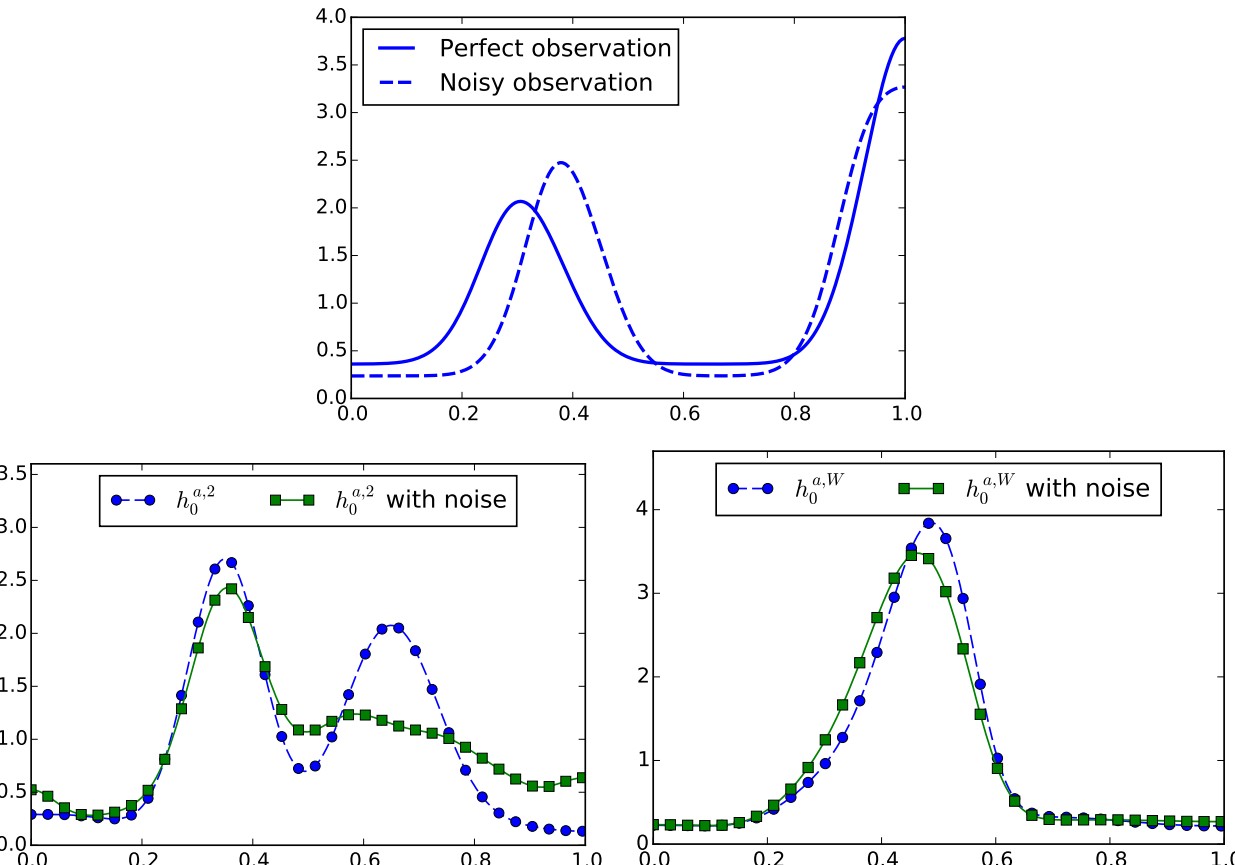

**Figure 6.** Top: Plot of an example of noise-free observations used in Section 4.2 experiment, equal to the true surface elevation $h^t$ at a given time. Plot of the corresponding observations with added noise, as described in Section 4.3. Bottom left: Analyses from the $\mathcal{L}^2$ cost function using perfect observations and observations with noise. Bottom right: Likewise with the Wasserstein cost function.

This example shows that Wasserstein cost function is more robust than $\mathcal{L}^2$ to such noise. This is quite a valuable feature for realistic applications.

## 5  Conclusions

We showed through some examples that, if not taken into account, position errors can lead to unrealistic initial conditions when using classical variational data assimilation methods. Indeed, such methods use the Euclidean distance which can behave poorly under position errors. To tackle this issue, we proposed instead the use of the Wasserstein distance to define the related cost function. The associated minimization algorithm was discussed and we showed that using descent iterations following Wasserstein geodesics lead to more consistent results.

On academic examples the corresponding cost function produces an analysis lying close to the Wasserstein average between the true and background states, and therefore has the same shape as them, and is well fit to correct position errors. This also gives more realistic predictions. This is a preliminary study, some issues have yet to be addressed for realistic applications, such as relaxing the constant-mass and positivity hypotheses and extending the problem to 2D applications.

5 Also, the interesting question of transposing this work into the filtering community (Kalman Filter, EnKF, particle filters, ...) raises the issue of writing a probabilistic interpretation of the Wasserstein cost function, which is out of our reach for now.

In particular the important theoretical aspect of representation of error statistics still requires to be thoroughly studied. Indeed classical implementations of variational data assimilation generally make use of $\mathcal{L}^2$ distances weighted by inverses of error covariance matrices. Analogy with Bayes formula allows for considering the minimization of the cost function as a 10 maximum likelihood estimation. Such an analogy is not straightforward with Wasserstein distances. Some possible research directions are given in Feyeux (2016) but this is beyond the scope of this paper. The ability to account for error statistics would also open the way for a proper use of the Wasserstein distance in Kalman-based data assimilation techniques.

**Appendix A: Proof of Theorem 3.1**

To prove Theorem 3.1, one first needs to differentiate the Wasserstein distance. The following Lemma from (Villani, 2003, 15 Theorem 8.13 p.264) gives the gradient of the Wasserstein distance.

**Lemma A.1** (Differentiation of the Wasserstein distance)**.** *Let $\rho_0, \rho_1 \in \mathcal{P}(\Omega)$, $\eta \in T_{\rho_0}\mathcal{P}$. For small enough $\epsilon \in \mathbb{R}$,*

$$\frac{1}{2}\mathcal{W}(\rho_0 + \epsilon\eta, \rho_1)^2 = \frac{1}{2}\mathcal{W}(\rho_0, \rho_1)^2 + \epsilon\langle\eta, \phi\rangle_2 + o(\epsilon) \tag{A1}$$

*with $\phi(x)$ the Kantorovich potential of the transport between $\rho_0$ and $\rho_1$.*

*Proof of Theorem 3.1.* Let $\rho_0 \in \mathcal{P}(\Omega)$ and $\eta = -\text{div}(\rho_0\nabla\Phi) \in T_{\rho_0}\mathcal{P}$. From the definition of $\mathcal{J}_\mathcal{W}$ in (16), from the defintion of 20 the tangent model (23) and in application of the Lemma A.1,

$$\lim_{\epsilon\to 0}\frac{\mathcal{J}_\mathcal{W}(\rho_0 + \epsilon\eta) - \mathcal{J}_\mathcal{W}(\rho_0)}{\epsilon} = \sum_{i=1}^{N^{obs}}\langle\mathbf{G}_i[\rho_0]\eta, \phi^i\rangle_2 + \omega_b\langle\eta, \phi^b\rangle_2$$

$$= \left\langle\eta, \sum_{i=1}^{N^{obs}}\mathbf{G}_i^*[\rho_0]\phi^i + \omega_b\phi^b\right\rangle_2$$

$$= \left\langle\eta, \sum_{i=1}^{N^{obs}}\mathbf{G}_i^*[\rho_0]\phi^i + \omega_b\phi^b + c\right\rangle_2 \tag{A2}$$

with $c$ such that the integral of the right hand side term is zero, so that the right hand side term belongs to $T_{\rho_0}\mathcal{P}$. The $\mathcal{L}^2$ gradient of $\mathcal{J}_\mathcal{W}$ is thus

$$\text{grad}_2\mathcal{J}_\mathcal{W}(\rho_0) = \sum_{i=1}^{N^{obs}}\mathbf{G}_i^*[\rho_0]\phi^i + \omega_b\phi^b + c \tag{A3}$$

To get the Wasserstein gradient of $\mathcal{J}_{\mathcal{W}}$, the same has to be done with the Wasserstein product. We let $\eta = -\mathrm{div}(\rho\nabla\Phi)$ and $g = \mathrm{grad}_2\mathcal{J}_{\mathcal{W}}(\rho_0)$ so that equations (A2) and (A3) give

$$
\begin{aligned}
\langle \eta, g \rangle_2 &= \langle -\mathrm{div}(\rho_0\nabla\Phi), g \rangle_2 \\
&= -\int_\Omega \mathrm{div}(\rho_0\nabla\Phi) g \\
&= \int_\Omega \rho_0 \nabla\Phi \nabla g.
\end{aligned}
\tag{A4}
$$

Last equality comes from Stokes theorem and from the fact that $\Phi$ is of zero normal derivative at the boundary. The last term gives the Wasserstein gradient because if $g$ is with Neumann boundary conditions, we have

$$
\int_\Omega \rho_0 \nabla\Phi \nabla g = \langle \eta, -\mathrm{div}(\rho_0\nabla g) \rangle_W,
\tag{A5}
$$

hence

$$
\forall \eta \in T_{\rho_0}\mathcal{P}, \quad \lim_{\epsilon\to 0} \frac{\mathcal{J}_{\mathcal{W}}(\rho_0 + \epsilon\eta) - \mathcal{J}_{\mathcal{W}}(\rho_0)}{\epsilon} = \langle \eta, -\mathrm{div}(\rho_0\nabla g) \rangle_W.
\tag{A6}
$$

$\square$

*Competing interests.* No competing interest are present.

*Acknowledgements.* The authors would like to thank the anonymous reviewers and the editor, whose comments helped to improve the paper, and C. Eldred for his editing. Nelson Feyeux is supported by the Région Rhône Alpes Auvergne through the ARC3 *Environment* PhD fellowship program.

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
