# Peer review of "Optimal transport for variational data assimilation"

_Nonlinear Processes in Geophysics, 2017_

## Referee Comment (RC1) · Anonymous Referee #1 · 15 Sep 2017

Review of *Optimal Transport for Variational Data Assimilation*
by Nelson Feyeux, Arthur Vidard and Maëlle Nodet

Recommendation :

Minor revision

General comments :

The aim of the paper is to introduce an alternative to the Euclidian distance emploied in variational formulation of data assimilation: the Wasserstein distance. The Wasserstein distance originates from the optimal transport and provides a better solution to the problem of phase error. The manuscript introduces this new metric and its use in DA at a theoretical level. Then numerical illustrations are provided in one dimension for a linear advection problem and for a nonlinear shallow water. This method relies on the restrictive assumption that part of the fields are "probability distribution" over compact support in geographical domain of physical space. Such "probability distribution" should not be confused by the classical probability distributions encountered in DA that represent the uncertainty in state space e.g. the forecast error distribution or the analysis error distribution.

The manuscript is well organized with an appropriate balance between the theoretical presentation from optimal-transport background and the numerical illustrations. However, it can be improved to facilitate its reading following the recommandations made in major comments.

Major comments :
1) The example introduced in Fig. 1 to illustrate the potential of the method is not clear and could be improve as follows:
a) You should precise the distribution name within the paragraph: "This is illustrated in Fig. 1 which shoes two densities $\rho_0$ and $\rho_1$. The second density $\rho_1$ can be seen as the first one $\rho_0$ with position error." ;
b) I guess the terminology of density & distribution & probability distribution should be avoid to prevent from any confusion in DA application, and especially the probabilistic interpretation of DA  (see next comments 2) );
c) You should introduce the formalism for $L^2$ cost functions saying that the minimum of the cost function $\|\rho - \rho_0\|^2_2 + \|\rho-\rho_1\|^2_2$ is given by $\rho_* = \frac{1}{2}(\rho_0 + \rho_1)$ ; while the average in the sens of the Wasserstein distance is the one of the figure, that is in between the two densities – without detailing the Wasserstein distance, as it is in the present manuscript.
2) The work presented here is limited to the case where the state vector and observations are positive fields with finite and normalized integral – part of the state vector is assume to be a probability measure over the domain – this seems very restrictive compared with the diversity of fields usually considered in data assimilation but solution to manage this issue can be considered (especially for image data). However the restriction to beeing a probability measure is not my objection: the problem I see is the possible confusion between probability distribution of error (forecast and analysis error distributions) and the particular case where a field (or part of the state vector) is a probability distribution. I think it would help the reader to insist on the difference between the classical framework of DA (with generic vector state) and this particular case, so to avoid any confusion between the particular field property (probability in compact domain in the physical space) and classical

error distribution (probability in state space): while mathematically appropriate, I think the terminology of probability densities P(Omega) (section2.2.1 and definition 2.1) should be replaced by something far from "probability densities". For instance in place of "probability densities" (title section 2.2.1 & definition), you could introduce a particular class for the fields, for instance it could be called "mass-class", keeping this terminology all along the manuscript, with a remark paragraph that would precise that in optimal transport what is socalled mass-class is actually probability distribution, indicating that the terminology is introduced to prevent from confusion with state/error probability distribution.

3) Kantorovitch potential (K-potential) plays a crucial role in the theoretical presentation as well as in the numerical solution of the minimizing process, but very few is said about its computation.
   - How the K- potential is it computed in this study : please give the detail of the algorithm used here, the indication provided in the manuscript about the construction of the K-potential in 1D (line 1-6 p6) is not enough. Detail, at least within a paragraph, how the K-potential can be computed in 2D/3D, even if only 1D example are considered here.
   - Illustrate what is the K-potential for the particular case of two gaussian distribution where $\rho_0$ ($\rho_1$) is a Gaussian of mean $m_0$ ($m_1$) and variance $\sigma_0^2$ ($\sigma_1^2$). If it exists, give the analytical expression for the potential in this case ?

4) p12,l1-2 and l14-15: Following the author and the numerical example developped in this section, the minimizing problem Eq(14) leads to two different solutions depending the choice of the dot product used along the minimizing process, but no detail is given explaining why this situation occurs. This could be due to possible mutliple minima of the cost function or to a non-convergence of the minimizing process when using the $L^2$ dot product. Authors mentioned the "success of the minimization of J_w" (l15) but without clearly indicating if the convergence was successful, or not, for the $L^2$ dot product. In this simple example, uniqueness of the minimum should be guaranted, indicating that the $L^2$ dot-product is not able to provide a good path toward the minimum. If this is correct than the author should mention it more clearly:
   "In this example, the minimizing process based on the $L^2$ scalar product fails to reach the unique minimum of the cost function as shown on … (additional illustration)"
   An additionnal figure (or panel in Fig.3) is needed to observe the non-convergence toward the minimum for this situation: please shows the value of the cost function J_w along the iterations of the minimizing process when using the two dot-products.
   I think a discussion is missing concerning existence and unicity of the J_w cost function, this should be included at the end of section 3.1.
   Is it possible to replace the steppest descent by a conjugated gradient ? Do you think that this replacement could improve the convergence for the L2 gradient ?

Minor comments:

1) p1, l11: "To achieve that goal" → "... this goal"
2) p1,l17: ".. to be sought (the control vector) is .." → ".. to be sought, the control vector, is .."
3) p7, l9: $\omega_b$ is not defined in Eq(13)
4) p3,l8: "Wasserstein distance is to compare" → " Wasserstein distance to compare"
5) p3,l9: "data assimilation Actual" → "data assimilation. Actual"
6) p3, l32: Observational operator is denoted by "G" in place of the more classical "H" notation. Please replace G into H along the manuscript.
7) P5,l23: Precise the page/section number in Ambrosio et al. (2008).
8) p 10, l14-18: Remind the equation number associated with the cost function and gradient. $L^2$ cost function is related with Eq.(2), Wasserstein cost function with Eq.(14), and the

iteration steps are deduced from Eq.(18).

9) P9, l17: write the push-forward for a given $x\in\Omega$ as $\rho_1[T(x)] |\det\nabla T_x| = \rho_0(x) $.

10) P10, l19: "$\alpha^n$ is chosen as optimal": explain how it is computed, and provide an appropriate reference.

---

## Referee Comment (RC2) · Anonymous Referee #2 · 19 Sep 2017

This is a very interesting paper, which introduces the use of optimal transport, and its metrics, in the world of data assimilation. The layout of the paper is very clear and appealing. For these reasons I strongly recommend the publication of this paper. However, the manuscript could benefit from the following remarks and suggestions:

- The English could and should be significantly improved.

- Clarifications are needed now and then, especially to help the reader who has little acquaintance with optimal transport.

- There is a bit of a contradiction in the willingness to introduce, or not, the transference plan view on optimal transport. This should be clarified.

- The authors could get rid of the mathematical remark style. In my opinion, it is not suited for NPG and is detrimental to the clarity of text. For the present manuscript, all the remarks can easily be naturally embedded in the text.

- For the sake of clarity, you should precisely define $\mathbf{y}_i^{obs}$ and $\mathbf{x}_0^b$ as mathematical objects.

- A recurrent question in data assimilation, which I believe many of your readers will have is: is there any probabilistic interpretation of the cost function defined with the Wasserstein distance? This is worth discussing it briefly.

- The paper would benefit from a more detailed discussion of the experiments, possibly another one with noise in the observation. There is room for it.

Specific remarks, in connection, or not, to the previous remarks are:

1. Title: why the capital letters in the title?

2. p.1, l.5: "With appropriate choices...": of what? Unclear.

3. p.1, l.6: "Optimal-transport-based optimization..." $\longrightarrow$ "Optimal transport-based optimization..."

4. p.1, l.6-7: "...to preserve the geometrical properties of the estimated initial condition.": this statement is too mysterious for an abstract. You should be more explicit.

5. p.1, l.13: "to use so-called data assimilation methods" $\longrightarrow$ "to use the so-called data assimilation methods"

6. p.1, l.14: It is uncanny that the third author is reluctant to cite her own brand new book on data assimilation.

7. p.1, l.14-15: "They aim at finding either the initial/boundary conditions or some parameters of a numerical model.": not only! They can be used for parameter estimation, reanalysis, etc.

8. p.1, l.18: "comparison between the observations and their model counterparts.": a mathematical expression called the innovation in data assimilation.

9. p.1, l.19: "unperfect" ⟶ "imperfect"

10. p.1, l.22: "More recently an hybrid of both approaches..." ⟶ "More recently hybrids of both approaches..."

11. p.2, l.1-2: "model counterparts. A Tikhonov regularization is also added and so the distance between the control vector and a background state carrying the a priori information is added in the cost function.": needs to be rephrased. It could be instead: "A Tikhonov regularization term is also added to the cost function as a distance between the control vector and a background state carrying the a priori information."

12. p.2, l.4: "aims to reach" ⟶ "aims at reaching"

13. p.2, l.4: "are smallest as possible." ⟶ "are as small as possible."

14. p.2, l.13: "...the desired localization." ⟶ "...the desired location."

15. p.2, l.17: "...has been founded by Monge..." ⟶ "...has been pioneered by Monge..."

16. p.2, l.19: I would remove "quickly".

17. p.3, l.1: "from pure mathematical analysis" ⟶ "from pure mathematical analysis on Riemannian spaces"

[Figure]

18. p.3, l.8: "...Wasserstein distance is to compare..." ⟶ "...Wasserstein distance to compare..."

19. p.3, l.9: "data assimilation Actual use of optimal transport" ⟶ "data assimilation. Actual use of optimal transport". Better, you could start a new paragraph with "Actual use...".

20. p.3, l.15: "This particularly subtle mathematical consideration is indeed crucial for the algorithm..." ⟶ "This particularly subtle mathematical considerations are indeed crucial for the algorithm..."

21. p.3, l.18: "...methods but it largely exceeds..." ⟶ "...methods, which largely exceeds..."

22. p.3, l.21: "required for the sequel" ⟶ "required in the following"

23. p.3, l.23-24: "Section 4 numerical illustrations are presented, choices for the gradients and the optimization methods are compared.": could be improved. Please rephrase.

24. p.3, l.24: "...and solutions proposed." ⟶ "...and solutions will be proposed."; the ellipsis could be avoided here.

25. p.3, l.26: "The section..." ⟶ "This section..."

26. p.3, l.27: "...materials...": principles?, facts?, properties?

27. p.3, l.28: "...production." ⟶ "...contribution."

28. p.4: You could mention that the Euclidean distances are local metrics, as opposed to the Wasserstein distance.

29. p.4, l.6: "...term xb which contains..." ⟶ "...term xb, which contains..."

30. p.4, l.6: "The actual cost function then writes..." ⟶ "The actual cost function then reads..."

31. p.4, l.15: " [0, 1] " ⟶ "the interval [0, 1] " since the notation is not really universal.

32. p.5, section 2.2.2: explain that the time $t$ is fictitious, or you will puzzle many readers.

33. p.5, l.9-11: Actually, I don't believe this is a necessary condition. There could non-zero fluxes of probability with a global balanced budget; see for instance Farchi et al. (2016).

34. p.5, l.18: Use \citep for the citation to Benamou and Brenier (2000).

35. p.5, l.21: "A remarkable point..." ⟶ "A remarkable property..."

36. p.5, l. 22: Use \citep for the citation to Ambrosio et al. (2008).

37. p.6, l.4-5: "...like the primal-dual Papadakis et al. (2014) or the semi-discrete Mérigot (2011).": I would be thrilled in meeting the primal dual Papadakis or discussing with the semi-discrete Mérigot... Please rephrase.

38. p.6, l.7: "... the scalar product choice conditions the gradient value." ⟶ "... the scalar product choice is used to define the gradient value."

39. p.6, l.11: "...shall formally be defined by..." ⟶ "...is formally defined by..."

40. p.6, l.11: "(cf. Otto (2001))": use \citep[][]{}.

41. p.6, Eq.(9): you probably should mention the set to which the Kantorovitch potential belongs.

42. p.6, l.15: This is not a proper sentence; you could merge it with the previous one.

43. p.6, l.23: "First we will consider..." $\longrightarrow$ "First, we will consider..."

44. p.6, l.24: "Second we will investigate..." $\longrightarrow$ "Second, we will investigate..."

45. p.6, l.24: "...we will investigate the role of the scalar product choice as well as the gradient descent method..." $\longrightarrow$ "...we will discuss the choice of the scalar product as well as the choice of the gradient descent method..."

46. p.7, l.5-6: another example, more accessible to the NPG readership, is the distance built in Farchi et al. (2016).

47. p.7, l.11: "...belonging respectively to $\mathcal{P}(\Omega)$ and $\mathcal{P}(\Omega_0)$." $\longrightarrow$ "...belonging to $\mathcal{P}(\Omega)$ and $\mathcal{P}(\Omega_0)$, respectively."

48. p.7, l.19: The scalar product is not unique (and as a consequence the gradient), but there is a natural one induced by the norm used in the cost function (here Wasserstein's). This could be mentioned, as the statement could be slightly puzzling for the reader.

49. p.7, l.26: It is not clear at this stage why you would use the $\mathcal{L}^2$ inner product.

50. p.8, l.24-25: Two "thus" in a row.

51. p.9, l.19: "...we will use after." $\longrightarrow$ "...we will use in the following."

52. p.9, l.14-24: You mentioned p.5, l.18-19 that the definition of optimal transport based on transference map is out of scope; and I am fine with it. I even think it was a clever choice. But, here, you finally use it and that seems important. This is quite frustrating for the reader, especially those who have little knowledge on optimal transport.

53. p.10, l.5: ", but results are still satisfactory.": Please remove the statement. It does not make sense to give the conclusion beforehand.

54. p.10, l.8-13: Why not consider, in addition, a case with observation noise; you perturb the Gaussian parameters of the observation, which would be similar to some bias in satellite observation.

55. p.10, l.19: "is chosen a optimal": vague, please be more specific.

56. p.11, Eq.(26): I would explicitly write the wind field in the equation even if it is uniformly equal to 1.

57. p.12, l.4: "...gaussians..." $\longrightarrow$ "...Gaussians..."

58. p.12, l.1: "The analyses $\rho_0^{a,W,2}$ and $\rho_0^{a,W,\#}$ are different even if they arise from the same cost function $\mathcal{J}_\mathcal{W}$, which highlights the need for a well-suited scalar-product.": that is one of the most interesting point of the experiment, but your comment is too short. You must elaborate. One would expect the numerical solutions to be the same, right? unless there is a convergence issue, which much be analysed and discussed and would fit nicely with what was laid in section 3.2.

59. p.12, l.18: "Shallow-Water" $\longrightarrow$ "shallow-water"

60. p.12, l.26: "Thanks to the wisdom gained..." $\longrightarrow$ "Thanks to the experience gained...": My wisdom told me that norm-induced scalar product was the best one from the very beginning.

61. p.13, Figure.4: please plot the observations, like you did for the first experiment.

62. p.13, l.9: "...badly..." $\longrightarrow$ "...poorly..."

63. p.12, l.12: Please avoid inverting subject and verb as this is much less frequent in English than in French.

**References**

Ambrosio, L., Gigli, N., Savaré, G., 2008. Gradient flows: in metric spaces and in the space of probability measures. Springer Science & Business Media.

Benamou, J.D., Brenier, Y., 2000. A computational fluid mechanics solution to the Monge-Kantorovich mass transfer problem. Numerische Mathematik 84, 375–393.

Farchi, A., Bocquet, M., Roustan, Y., Mathieu, A., Quérel, A., 2016. Using the Wasserstein distance to compare fields of pollutants: application to the radionuclide atmospheric dispersion of the Fukushima-Daiichi accident. Tellus B 68, 31682. doi:`10.3402/tellusb.v68.31682`.

---

## Editor Comment (EC1) · O. Talagrand (Editor) · 25 Sep 2017

Both referees have recommended acceptance of the paper subject to minor revisions. As Editor I suggest that the authors (if they have not already done so) start writing a revised version of their paper, taking into account the comments of the referees. They may of course also submit to the interactive discussion any response they may have to the referees' comments.

I make in addition the following comments.

1. Concerning Figure 3 (top right) and the fact that the two inner products lead to distinct 'minima' of the cost function (13) (see major comment 4 of Referee 1 and specific remark 58 of Referee 2), I note that the steepest gradient algorithm is known to be very inefficient. The 'failure' of the $\mathcal{L}^2$ gradient may be therefore due as much to the choice of the descent algorithm as to the choice of the inner product. As suggested by Referee 1, replacing the steepest descent algorithm by another algorithm, such as a conjugate gradient one, might be useful.

2. As noted by Referee 2, the English of the paper needs significant improvement. The Referee makes quite a few suggestions, and, once a paper has been accepted for publication, *Nonlinear Processes in Geophysics* provides free copy-editing, intended primarily at correcting the English if necessary. It would however be preferable that the authors have their paper checked by a native English speaker.

Other comments.

3. It does not seem to be said, in either one of the numerical applications, what the dimension of the discretized control space is. And it does not seem to be said what $\Omega$ is subsection 4.2 (*Non-linear example*).

4. Eq. (6). Most readers of *NPG* will not be familiar with the Wasserstein metric. It might be useful to explain the significance of the indices 2 in $\mathcal{W}^2_2$ in (or to remove them since they are not useful for the paper anyway).

5. I understand Eq. (9) defines $T_{\rho 0}\mathcal{P}$ as the set of potentials $\Phi$ that verify the conditions on the right-hand side of the equation. Say it clearly (see also specific remark 41 of Referee 2).

6. P. 8, l. 21, symbols $= 0$ missing (see l. 12 higher up).

7. Figure 4, end of caption … *at the output of the model* $\rightarrow$ … *at the end of the assimilation window*.

---

## Author Comment (AC1) · 13 Nov 2017

*a revised version of the paper is attached as a supplement*
Reply to referee 1 (RC1)

We would like to thank the referee for his/her extensive review on our paper and for giving us the opportunity to improve our paper.
We copied your commentary in italics below, we reply in normal font.

*Major comments :*

*1) The example introduced in Fig. 1 to illustrate the potential of the method is not clear and could be improve as follows: a) You should precise the distribution name within*

[Figure]

the paragraph: *"This is illustrated in Fig. 1 which shoes two densities $\rho_0$ and $\rho_1$. The second density $\rho_1$ can be seen as the first one $\rho_0$ with position error."* ;

Corrected, thank you.

*b) I guess the terminology of density & distribution & probability distribution should be avoid to prevent from any confusion in DA application, and especially the probabilistic interpretation of DA (see next comments 2) );*

Ok. In the introduction, we replaced "density" with either "curve" (to describe $\rho_0$ and $\rho_1$) or "measure" or "mass" (in the optimal transport section). See comments 2 for the rest of the paper.

*c) You should introduce the formalism for $L^2$ cost functions saying that the minimum of the cost function $||\rho - \rho_0||_2^2 + ||\rho - \rho_1||_2^2$ is given by $\rho_* = \frac{1}{2}(\rho_0 + \rho_1)$ ; while the average in the sens of the Wasserstein distance is the one of the figure, that is in between the two densities – without detailing the Wasserstein distance, as it is in the present manuscript.*

Ok.

*2) The work presented here is limited to the case where the state vector and observations are positive fields with finite and normalised integral – part of the state vector is assume to be a probability measure over the domain – this seems very restrictive compared with the diversity of fields usually considered in data assimilation but solution to manage this issue can be considered (especially for image data). However the restriction to being a probability measure is not my objection: the problem I see is the possible confusion between probability distribution of error (forecast and analysis error distributions) and the particular case where a field (or part of the state vector)*

*is a probability distribution. I think it would help the reader to insist on the difference between the classical framework of DA (with generic vector state) and this particular case, so to avoid any confusion between the particular field property (probability in compact domain in the physical space) and classical error distribution (probability in state space): while mathematically appropriate, I think the terminology of probability densities P(Omega) (section2.2.1 and definition 2.1) should be replaced by something far from "probability densities". For instance in place of "probability densities" (title section 2.2.1 & definition), you could introduce a particular class for the fields, for instance it could be called "mass-class", keeping this terminology all along the manuscript, with a remark paragraph that would precise that in optimal transport what is so-called mass-class is actually probability distribution, indicating that the terminology is introduced to prevent from confusion with state/error probability distribution.*

Ok, thank you. We removed all occurrences of "densities", we replaced them by "mass functions". Following your suggestion, we included a remark in paragraph 2.2.1 Mass functions (previously "probability densities").

*3) Kantorovitch potential (K-potential) plays a crucial role in the theoretical presentation as well as in the numerical solution of the minimising process, but very few is said about its computation. - How the K- potential is it computed in this study : please give the detail of the algorithm used here, the indication provided in the manuscript about the construction of the K- potential in 1D (line 1-6 p6) is not enough. Detail, at least within a paragraph, how the K- potential can be computed in 2D/3D, even if only 1D example are considered here.*

Ok, we added such a paragraph detailing numerical computation of K and W2 in 1D and 2/3D, at the end of 2.2.2.

*- Illustrate what is the K-potential for the particular case of two gaussian distribution*

*where $\rho_0$ ($\rho_1$) is a Gaussian of mean $m_0$ ($m_1$) and variance $\sigma_0^2$ ($\sigma_1^2$). If it exists, give the analytical expression for the potential in this case ?*

Ok. We included such an example (Example 2.3) at the end of Section 2.2.2.

*4) p12,l1-2 and l14-15: Following the author and the numerical example developed in this section, the minimising problem Eq(14) leads to two different solutions depending the choice of the dot product used along the minimising process, but no detail is given explaining why this situation occurs. This could be due to possible multiple minima of the cost function or to a non-convergence of the minimising process when using the $L^2$ dot product. Authors mentioned the "success of the minimisation of $J_w''$ (l15) but without clearly indicating if the convergence was successful, or not, for the $L^2$ dot product. In this simple example, uniqueness of the minimum should be guaranteed, indicating that the $L^2$ dot-product is not able to provide a good path toward the minimum. If this is correct than the author should mention it more clearly: "In this example, the minimising process based on the $L^2$ scalar product fails to reach the unique minimum of the cost function as shown on ... (additional illustration)"*

*An additional figure (or panel in Fig.3) is needed to observe the non-convergence toward the minimum for this situation: please shows the value of the cost function $J_w$ along the iterations of the minimising process when using the two dot-products.*

*I think a discussion is missing concerning existence and unicity of the $J_w$ cost function, this should be included at the end of section 3.1.*

*Is it possible to replace the steepest descent by a conjugated gradient ? Do you think that this replacement could improve the convergence for the $L^2$ gradient ?*

Ok, thank you for this helpful comment.

Regarding unicity of Jw's minimiser, we added a few sentences at the end of Section 3.1 (page 8).

Regarding conjugate gradient, we added a plot in Figure 3 comparing convergence speed of (DG#), (DG2) and a version of (DG2) using the conjugate gradient algorithm. Conjugate gradient speeds up the algorithm, but is not as fast as (DG#). See Figure 3 (page 13) and the third paragraph of 4.1 (page 12) about it.

*Minor comments:*

*1) p1, l11: "To achieve that goal" → "... this goal"* **Ok**

*2) p1,l17: ".. to be sought (the control vector) is .." → ".. to be sought, the control vector, is .."* **Ok**

*3) p7, l9: $\omega_b$ is not defined in Eq(13)* **Ok**

*4) p3,l8: "Wasserstein distance is to compare" → " Wasserstein distance to compare"* **Ok**

*5) p3,l9: "data assimilation Actual" → "data assimilation. Actual"* **Ok**

*6) p3, l32: Observational operator is denoted by "G" in place of the more classical "H" notation. Please replace G into H along the manuscript.*

Actually in our manuscript, G denote $H \circ M$, and it is a classical notation in DA. However, our phrasing was indeed unfit in Section 2.1, so we clarified: our control vector is $x_0$ the system initial state, and not $x$ as we wrote in the first version of our paper. All occurrences of $x$ have been replaced accordingly in Section 2.1, so that the use of $\mathcal{G}$ is now fit.

*7) P5,l23: Precise the page/section number in Ambrosio et al. (2008).*

It is more easily accessible in Benamou and Brenier, 2000, so we actually changed the reference.

*8) p 10, l14-18: Remind the equation number associated with the cost function and gradient. $L^2$ cost function is related with Eq.(2), Wasserstein cost function with Eq.(14), and the iteration steps are deduced from Eq.(18).* **Ok**

*9) P9,l17:write the push-forward for a given $x \in \Omega$ as $\rho_1[T(x)]|det\nabla T_x| = \rho_0(x)$.*

This remark has been removed, following Referee's 2 comment. See Section 3.3 where the Monge-Ampere terminology of OT (with a transport map $T$) has been removed to only deal with the Benamou and Brenier formulation (with $v$).

*10) P10, l19: "$\alpha^n$ is chosen as optimal": explain how it is computed, and provide an appropriate reference.*

We specified that $\alpha^n$ is found using a line search algorithm. It is therefore not strictly optimal but approximately optimal.

**Supplement:**

[revised manuscript text omitted]

---

## Author Comment (AC2) · 13 Nov 2017

*a revised version of the paper is attached as a supplement*
Reply to Referee 2

We would like to thank the referee for his/her extensive review on our paper and for giving us the opportunity to improve our paper.
We copied your commentary in italics below, we reply in normal font.

*This is a very interesting paper, which introduces the use of optimal transport, and its metrics, in the world of data assimilation. The layout of the paper is very clear and appealing. For these reasons I strongly recommend the publication of this paper.*

[Figure]

*However, the manuscript could benefit from the following remarks and suggestions:*

*• The English could and should be significantly improved.*

The paper has been read and corrected by a native English speaker.

*• Clarifications are needed now and then, especially to help the reader who has little acquaintance with optimal transport.*

Ok.

*• There is a bit of a contradiction in the willingness to introduce, or not, the transference plan view on optimal transport. This should be clarified.*

Ok.

*• The authors could get rid of the mathematical remark style. In my opinion, it is not suited for NPG and is detrimental to the clarity of text. For the present manuscript, all the remarks can easily be naturally embedded in the text.*

Done.

*• For the sake of clarity, you should precisely define yobs and xb as mathematical objects.*

It is done between equations (14) and (15).

*• A recurrent question in data assimilation, which I believe many of your readers will have is: is there any probabilistic interpretation of the cost function defined with the Wasserstein distance? This is worth discussing it briefly.*

It is a very interesting question indeed, but we do not have any clear answer yet. We added a line on this topic in the conclusion.

*• The paper would benefit from a more detailed discussion of the experiments, possibly another one with noise in the observation. There is room for it.*

We added such experiment and a section commenting it

*Specific remarks, in connection, or not, to the previous remarks are:*

*1. Title: why the capital letters in the title?*

Out of habit in some journals... Corrected.

*2. p.1, l.5: "With appropriate choices...": of what? Unclear.*

*3. p.1, l.6: "Optimal-transport-based optimization..." → "Optimal transport-based optimization..."*

*4. p.1, l.6-7: "...to preserve the geometrical properties of the estimated initial condition.": this statement is too mysterious for an abstract. You should be more explicit.*

Indeed that was cryptic. The abstract has been rewritten.

*5. p.1, l.13: "to use so-called data assimilation methods" → "to use the so-called data assimilation methods"* Ok

*6. p.1, l.14: It is uncanny that the third author is reluctant to cite her own brand new book on data assimilation.* :-) Done, thanks

*7. p.1, l.14-15: "They aim at finding either the initial/boundary conditions or some parameters of a numerical model.": not only! They can be used for parameter estimation, reanalysis, etc.* Of course. Corrected.

*8. p.1, l.18: "comparison between the observations and their model counterparts.": a mathematical expression called the innovation in data assimilation.* Yes, but actually the innovation is defined as the subtraction of the obs and their model counterparts, and here we talk about a comparison, which is not necessarily a subtraction. We tried to introduce the innovation here but it causes a problem later on because we really talk about distances between obs and model, and not only "norm of the innovation". So we decided with statu quo here.

*9. p.1, l.19: "unperfect" → "imperfect"* Ok

*10. p.1, l.22: "More recently an hybrid of both approaches..." → "More recently hybrids of both approaches..."* Ok

*11. p.2, l.1-2: "model counterparts. A Tikhonov regularization is also added and so the distance between the control vector and a background state carrying the a priori information is added in the cost function.": needs to be rephrased. It could be instead: "A Tikhonov regularization term is also added to the cost function as a distance between the control vector and a background state carrying the a priori information."* Ok

*12. p.2, l.4: "aims to reach" → "aims at reaching"* Ok

*13. p.2, l.4: "are smallest as possible." → "are as small as possible."* Ok

*14. p.2, l.13: "...the desired localization." → "...the desired location."* Ok

*15. p.2, l.17: "...has been founded by Monge..." → "...has been pioneered by Monge..."* Ok

*16. p.2, l.19: I would remove "quickly".* Ok

*17. p.3, l.1: "from pure mathematical analysis" → "from pure mathematical analysis on Riemannian spaces"* Ok

*18. p.3, l.8: "...Wasserstein distance is to compare..." → "...Wasserstein distance to compare..."* Ok

*19. p.3, l.9: "data assimilation Actual use of optimal transport" → "data assimilation. Actual use of optimal transport". Better, you could start a new paragraph with "Actual use...".* Ok

*20. p.3, l.15: "This particularly subtle mathematical consideration is indeed crucial for the algorithm..." → "This particularly subtle mathematical considerations are indeed crucial for the algorithm..."* Ok

*21. p.3, l.18: "...methods but it largely exceeds..." → "...methods, which largely exceeds..."* Ok

*22. p.3, l.21: "required for the sequel"* → *"required in the following"* Ok

*23. p.3, l.23-24: "Section 4 numerical illustrations are presented, choices for the gradients and the optimization methods are compared.": could be improved. Please rephrase.* Ok

*24. p.3, l.24: "...and solutions proposed."* → *"...and solutions will be proposed."; the ellipsis could be avoided here.* Ok

*25. p.3, l.26: "The section..."* → *"This section..."* Ok

*26. p.3, l.27: "...materials...": principles?, facts?, properties?* Ok: concepts, method, principles, main theorems.

*27. p.3, l.28: "...production."* → *"...contribution."* Ok

*28. p.4: You could mention that the Euclidean distances are local metrics, as opposed to the Wasserstein distance.* Ok, at the end of paragraph 2.1, just before we introduce the W2 distance.

*29. p.4, l.6: "...term xb which contains..."* → *"...term xb, which contains..."* Ok

*30. p.4, l.6: "The actual cost function then writes..."* → *"The actual cost function then reads..."* Ok

*31. p.4, l.15: " [0, 1] " → "the interval [0, 1] " since the notation is not really universal.* Ok

*32. p.5, section 2.2.2: explain that the time t is fictitious, or you will puzzle many readers.* Thanks! We added two sentences between eqs (5) and (6).

*33. p.5, l.9-11: Actually, I don't believe this is a necessary condition. There could non-zero fluxes of probability with a global balanced budget; see for instance Farchi et al. (2016).* Thanks, it is indeed a sufficient condition, corrected.

*34. p.5, l.18: Use "citep" for the citation to Benamou and Brenier (2000).* Ok

*35. p.5, l.21: "A remarkable point..." → "A remarkable property..."* Ok

*36. p.5, l. 22: Use citep for the citation to Ambrosio et al. (2008).* Ok

*37. p.6, l.4-5: "...like the primal-dual Papadakis et al. (2014) or the semi-discrete MeÌĄrigot (2011).": I would be thrilled in meeting the primal dual Papadakis or discussing with the semi-discrete MeÌĄrigot. . . Please rephrase.*

We used citep instead.

*38. p.6, l.7: "... the scalar product choice conditions the gradient value." → "... the scalar product choice is used to define the gradient value."* Ok

*39. p.6, l.11: "...shall formally be defined by..." → "...is formally defined by..."* Ok
*40. p.6, l.11: "(cf. Otto (2001))": use citep[][].* Ok

*41. p.6, Eq.(9): you probably should mention the set to which the Kantorovitch potential belongs.*

This is now eq (10): we started the set with $\{\eta \in L^2, \text{ s.t. } \eta = -\text{div}(\rho_0 \nabla \Phi)....$ We do not say more precisely where is $\Phi$ because $\Phi$ could be very general, its only requirement is to be so that $-\text{div}(\rho_0 \nabla \Phi)$ exists. As it would make the definition quite heavy we thought it best not to say anything about $\Phi$. It's also quite complex to get information in the main litterature (Villani e.g.), we simply do not know what is, in general, the nature of $\Phi$.

*42. p.6, l.15: This is not a proper sentence; you could merge it with the previous one.* Ok

*43. p.6, l.23: "First we will consider..."* → *"First, we will consider..."* Ok

*44. p.6, l.24: "Second we will investigate..."* → *"Second, we will investigate..."* Ok

*45. p.6, l.24: "...we will investigate the role of the scalar product choice as well as the gradient descent method..."* → *"...we will discuss the choice of the scalar product as well as the choice of the gradient descent method..."* Ok

*46. p.7, l.5-6: another example, more accessible to the NPG readership, is the distance built in Farchi et al. (2016).* Ok

*47. p.7, l.11: "...belonging respectively to P($\Omega$) and P($\Omega$0)."* → *"...belonging to P($\Omega$) and P($\Omega$0), respectively."* Ok

*48. p.7, l.19: The scalar product is not unique (and as a consequence the gradient), but there is a natural one induced by the norm used in the cost function (here Wasserstein's). This could be mentioned, as the statement could be slightly puzzling for the reader.*

Ok, we clarified at the beginning of 3.2.

*49. p.7, l.26: It is not clear at this stage why you would use the L2 inner product.*

Ok, we clarified at the beginning of 3.2.

*50. p.8, l.24-25: Two "thus" in a row.* Ok

*51. p.9, l.19: "...we will use after." → "...we will use in the following."* Ok

*52. p.9, l.14-24: You mentioned p.5, l.18-19 that the definition of optimal transport based on transference map is out of scope; and I am fine with it. I even think it was a clever choice. But, here, you finally use it and that seems important. This is quite frustrating for the reader, especially those who have little knowledge on optimal transport.*

Ok, we changed what was remark 3.4 and included a wide paragraph in Section 3.3 to explain the notation $\#$ as simply as we could, using geodesics.

*53. p.10, l.5: ", but results are still satisfactory.": Please remove the statement. It does not make sense to give the conclusion beforehand.* Ok

*54. p.10, l.8-13: Why not consider, in addition, a case with observation noise; you*

*perturb the Gaussian parameters of the observation, which would be similar to some bias in satellite observation.*

We added such an experiment (Section 4.3), it shows that the W2 distance is more robust to this type of noise than L2.

*55. p.10, l.19: "is chosen a optimal": vague, please be more specific.*

Line search, we clarified the text.

*56. p.11, Eq.(26): I would explicitly write the wind field in the equation even if it is uniformly equal to 1.* Ok

*57. p.12, l.4: "...gaussians..." → "...Gaussians..."* Ok

*58. p.12, l.1: "The analyses $\rho a, W, 2$ and $\rho a, W, \#$ are different even if they arise from the same cost function JW, which highlights the need for a well-suited scalar- product.": that is one of the most interesting point of the experiment, but your comment is too short. You must elaborate. One would expect the numerical solutions to be the same, right? unless there is a convergence issue, which much be analysed and discussed and would fit nicely with what was laid in section 3.2.*

Ok, a convergence figure has been added following referee1, and we discussed this point in Section 4.1 (+ added a few words about the minimizer uniqueness at the end of Section 3.1).

*59. p.12, l.18: "Shallow-Water" → "shallow-water"* Ok

*60. p.12, l.26: "Thanks to the wisdom gained..." → "Thanks to the experience gained...": My wisdom told me that norm-induced scalar product was the best one from the very beginning.* Ok

*61. p.13, Figure.4: please plot the observations, like you did for the first experiment.* Ok

*62. p.13, l.9: "...badly..." → "...poorly..."* Ok

*63. p.12, l.12: Please avoid inverting subject and verb as this is much less frequent in English than in French.*

Ok.

---

## Author Comment (AC3) · 13 Nov 2017

*A revised version of the paper has been added as a supplement*
Reply to the Editor

We would like to thank the editor for his review of our paper and for giving us the opportunity to improve our paper.
We copied your commentary in italics below, we reply in normal font.

*Both referees have recommended acceptance of the paper subject to minor revisions. As Editor I suggest that the authors (if they have not already done so) start writing a revised version of their paper, taking into account the comments of the referees. They*

[Figure]

*may of course also submit to the interactive discussion any response they may have to the referees' comments.*

*I make in addition the following comments.*

*1. Concerning Figure 3 (top right) and the fact that the two inner products lead to distinct 'minima' of the cost function (13) (see major comment 4 of Referee 1 and specific remark 58 of Referee 2), I note that the steepest gradient algorithm is known to be very inefficient. The 'failure' of the L2 gradient may be therefore due as much to the choice of the descent algorithm as to the choice of the inner product. As suggested by Referee 1, replacing the steepest descent algorithm by another algorithm, such as a conjugate gradient one, might be useful.*

A comparison of the steepest descent algorithm (DG2) with a conjugate gradient algorithm has been shown in the Figure 3. This conjugate gradient is faster but not as quick as (DG#). A note has been added in the third paragraph of 4.1.

*2. As noted by Referee 2, the English of the paper needs significant improvement. The Referee makes quite a few suggestions, and, once a paper has been accepted for publication, Nonlinear Processes in Geophysics provides free copy-editing, intended primarily at correcting the English if necessary. It would however be preferable that the authors have their paper checked by a native English speaker.*

Ok.

*Other comments.*

*3. It does not seem to be said, in either one of the numerical applications, what the dimension of the discretized control space is. And it does not seem to be said what $\Omega$ is subsection 4.2 (Non-linear example).*

Ok, dimension added in the introduction of Section 4. $\Omega$ is the same for both experiments, described at the beginning of Section 4. We added a sentence at the beginning of 4.2 to refer the reader to 4 for the experimental framework details.

*4. Eq. (6). Most readers of NPG will not be familiar with the Wasserstein metric. It might be useful to explain the significance of the indices 2 in W2 (or to remove them since they are not useful for the paper anyway).*

Ok, we removed them.

*5. I understand Eq. (9) defines $T_\rho P$ as the set of potentials $\Phi$ that verify the conditions on the right-hand side of the equation. Say it clearly (see also specific remark 41 of Referee 2).*

Ok. We clarified the definition, as the tangent space is actually the set of $\eta$, such that there exists $\Phi$ such that (...).

*6. P. 8, l. 21, symbols = 0 missing (see l. 12 higher up).*

Ok.

*7. Figure 4, end of caption ... at the output of the model → ... at the end of the assimilation window.*

Ok.

---

## Editor Decision (ED1)

Cher Arthur,

Two referees have now sent their evaluations of the revised version of your paper. They are the same as the referees of the previous version. Although both had asked only for minor revisions, and not for a new review, I wanted to know whether they were satisfied with the new version. Both have now recommended acceptance of the paper (I put below the comments of referee 2, to which I think you have no access).

As editor, I however consider further clarification is desirable before the paper can be published. In the line of a comment that was made by Referee 1 on the previous version of the paper (his/her major comment 2), it might be useful to stress more strongly that you are dealing with (positive) physical fields, and not probability distributions. For instance, you might state explicitly in the comments following eq. (4) that the 'mass functions' considered in the paper are physical fields. And I suggest you avoid the word 'gaussian' (which will be automatically associated with probability distributions in the minds of some readers), for instance in section 4 (*Numerical illustrations*). You may speak of *squared exponential*, with the word 'gaussian' in parentheses.

And two additional remarks.

- Both notations $\mathcal{L}^2$ and $L^2$ seem to be used indifferently (see, *e.g.*, ll. 22 and 27, p. 4). Use consistent notations.

- Caption of Fig. 3, l. 3, *outputs of the model* → *fields at final time*

I look forward to receiving the final version of your paper.

PS. Comments of referee 2

I have read: (i) the authors' response entirely (ii) the introduction and the conclusion (iii) a selection of paragraphs (iv) the new experiment section.

I am very satisfied with the authors' response and how they handled the most important points raised by both reviewers.

Moreover, from the extracts that I have read in the manuscript, the English has considerably improved.

Best regards,
XXXX

Two minor typos in the introduction that you may pass on to the authors:
Page 3, line 11: Bonneel et al. (2011) should use called by a \citep.
Page 3, line 26: "lays" should be "lies"

---

## Author Response (AR2)

Reply to the Editor, 1st comment

*As editor, I however consider one point needs clarification before the paper can be published. It is the experimental set-up of subsection 4.2 (Nonlinear example). You write that you have used the shallow-water equations (p. 14, l. 8). What is the domain for the space variable x (from what I understand, it cannot be the set , which is a range of possible values for the initial surface elevation h0(x)) ? And what is the connection between the field h0(x) and the scalar random variable for which you want to determine a probability distribution ? Maybe I have misunderstood something, but that point must be clarified.*

This point should be now clear, according to the 2nd comment. At the beginning of Section 4.2 we state that the framework is the same as the first test-case and we refer to 4.1 for details, where the reader can find, e.g., the definition $\Omega = [0, 1]$ and other information about the state variable.

*- Both notations $\mathcal{L}^2$ and $L^2$ seem to be used indifferently (see, e.g., ll. 22 and 27, p. 4). Use consistent notations.*
Ok, done.

*- Caption of Fig. 3, l. 3, outputs of the model fields at final time.*
Ok, done.

Reply to the Editor, 2nd comment

*Je viens de t'envoyer mes Editor's comments sur le papier Feyeux et al. Je me doutais bien qu'il y avait quelque chose que j'avais mal compris. Je comprends maintenant que les figures 5 et 6 representent bien des champs h(x), et non des distributions de probabilite. Peut-etre faut-il que ce soit dit un peu plus clairement.*

There was a mistake in the figure, we used rho instead of h. We corrected this and named the variable explicitly in the text.

*J'ai fondamentalement ete trompe par le fait que vous vous restreignez a des fonctions postives (eq. 4), ce qui est tres limitatif pour un un champ physique. Je suppose d'ailleurs que la distance de Wasserstein n'est definie que pour des champs positifs, ce qui necessite de considerer des champs ayant une borne inferieure stricte. Je suggere que vous le mentionniez.*
Ok, done

*Enfin, je suggere que vous evitiez l'utilisation du mot 'gaussien', qui ne peut guere dans l'esprit de beaucoup etre associe aux variations spatiales d'un champ physique.*
We replaced gaussian by "localised mass function" where we could

Reply to the 2nd Referee

*Two minor typos in the introduction that you may pass on to the authors: Page 3, line 11: Bonneel et al. (2011) should use called by a citep. Page 3, line 26: "lays" should be "lies"*

Corrected, thank you.